# Mutations in LRRK2 linked to Parkinson disease sequester Rab8a to damaged lysosomes and regulate transferrin-mediated iron uptake in microglia

**Adamantios Mamais**[1,2], **Jillian H. Kluss**[1], **Luis Bonet-Ponce**[1], **Natalie Landeck**[1], **Rebekah G. Langston**[1], **Nathan Smith**[3], **Alexandra Beilina**[1], **Alice Kaganovich**[1], **Manik C. Ghosh**[4], **Laura Pellegrini**[5], **Ravindran Kumaran**[1], **Ioannis Papazoglou**[6], **George R. Heaton**[1], **Rina Bandopadhyay**[7], **Nunziata Maio**[4], **Changyoun Kim**[8], **Matthew J. LaVoie**[2], **David C. Gershlick**[9], **Mark R. Cookson**[1]*

1 Cell Biology and Gene Expression Section, National Institute on Aging, National Institutes of Health, Maryland, United States of America, 2 Department of Neurology, University of Florida, Gainesville, Florida, United States of America, 3 Department of Biochemistry and the Redox Biology Center, University of Nebraska, Lincoln, Nebraska, United States of America, 4 Molecular Medicine Branch, 'Eunice Kennedy Shriver' National Institute of Child Health and Human Development, Bethesda, Maryland, United States of America, 5 MRC Laboratory of Molecular Biology, Cambridge, United Kingdom, 6 National Institute of Diabetes and Digestive and Kidney Diseases, National Institutes of Health, Bethesda, Maryland, United States of America, 7 UCL Institute of Neurology and Reta Lila Weston Institute of Neurological Studies, University College London, London, United Kingdom, 8 Molecular Neuropathology Section, Laboratory of Neurogenetics, National Institute on Aging, National Institutes of Health, Bethesda, Maryland, United States of America, 9 Cambridge Institute for Medical Research, University of Cambridge, Cambridge, United Kingdom

* cookson@mail.nih.gov

**Data Availability Statement:** All plotted data are included in supplementary file S1 Data. The bulk RNA Seq and single-cell RNA Seq data presented

## Abstract

Mutations in leucine-rich repeat kinase 2 (LRRK2) cause autosomal dominant Parkinson disease (PD), while polymorphic LRRK2 variants are associated with sporadic PD. PD-linked mutations increase LRRK2 kinase activity and induce neurotoxicity in vitro and in vivo. The small GTPase Rab8a is a LRRK2 kinase substrate and is involved in receptor-mediated recycling and endocytic trafficking of transferrin, but the effect of PD-linked LRRK2 mutations on the function of Rab8a is poorly understood. Here, we show that gain-of-function mutations in LRRK2 induce sequestration of endogenous Rab8a to lysosomes in overexpression cell models, while pharmacological inhibition of LRRK2 kinase activity reverses this phenotype. Furthermore, we show that LRRK2 mutations drive association of endocytosed transferrin with Rab8a-positive lysosomes. LRRK2 has been nominated as an integral part of cellular responses downstream of proinflammatory signals and is activated in microglia in postmortem PD tissue. Here, we show that iPSC-derived microglia from patients carrying the most common LRRK2 mutation, G2019S, mistraffic transferrin to lysosomes proximal to the nucleus in proinflammatory conditions. Furthermore, G2019S knock-in mice show a significant increase in iron deposition in microglia following intrastriatal LPS injection compared to wild-type mice, accompanied by striatal accumulation of ferritin. Our data support a role of LRRK2 in modulating iron uptake and storage in response to proinflammatory stimuli in microglia.

in Fig 5 have been deposited in NCBI's Gene
Expression Omnibus and are accessible through
GEO Series accession numbers GSE186483 and.
GSE186559 respectively (https://www.ncbi.nlm.
nih.gov/geo/query/acc.cgi?acc=GSE186483,
https://www.ncbi.nlm.nih.gov/geo/query/acc.cgi?
acc=GSE186559).

**Funding:** This work was supported in part by the
National Institutes of Health, NIA, Intramural
Research Program (MRC), by NIH extramural
grant NS110188 (MJL) and the NIA IRP
Postdoctoral grant scheme (AM). The funders had
no role in study design, data collection and
analysis, decision to publish, or preparation of the
manuscript.

**Competing interests:** The authors have declared
that no competing interests exist.

**Abbreviations:** ERC, endocytic recycling
compartment; ICP-MS, inductively coupled plasma
mass spectrometry; iPSC, induced pluripotent
stem cell; KO, knockout; LPS, lipopolysaccharide;
LRRK2, leucine-rich repeat kinase 2; NDS, Normal
Donkey Serum; PBMC, peripheral mononuclear
blood cell; PD, Parkinson disease; PPMI, Parkinson
Progression Marker Initiative; RT, room
temperature; scRNA-Seq, single-cell RNA-Seq;
siRNA, small interfering RNA; SN, substantia nigra;
TfR, transferrin receptor; WT, wild-type.

## Introduction

Missense mutations on the leucine-rich repeat kinase 2 (LRRK2) gene cause autosomal dominant Parkinson disease (PD), while common genetic variants identified by genome-wide association studies have been linked to idiopathic PD and inflammatory diseases [1–3]. LRRK2 is a multidomain enzyme, and PD-linked mutations are localized to its enzymatic domains, namely the kinase domain and Ras of complex proteins and C-terminal of ROC (ROC-COR) domains. LRRK2 has been linked to a number of cellular pathways including autophagy, lysosomal processing, inflammation, and vesicular trafficking [4]. LRRK2 is expressed in immune cells, and prior observations suggest a role of LRRK2 in microglial activation and an effect of PD-linked mutations on cytokine release and inflammation [5–7]. The kinase activity of LRRK2 is thought to drive disease pathology and thus is being targeted pharmacologically as a possible PD therapy [4].

A specific role of LRRK2 in vesicular trafficking has been nominated in recent years, and a number of studies have highlighted how downstream signaling pathways are affected by PD-linked mutations. LRRK2 can phosphorylate a subset of Rab GTPases, including Rab8a, Rab10, and Rab29, at a conserved motif, and this phosphorylation regulates Rab activity and association with effector molecules [8,9]. Rab GTPases control the spatiotemporal regulation of vesicle traffic and are involved in vesicle sorting, motility, and fusion [10]. Different Rab GTPases exhibit selectivity for different cellular compartments, thus conferring membrane identity that governs secretory and endocytic pathways [11]. For example, LRRK2 associates with Rab29, a Rab GTPase that is involved in vesicular transport and protein sorting [12,13] and a candidate risk gene for sporadic PD [14]. Rab29 can mediate recruitment of LRRK2 to the *trans*-Golgi network, inducing LRRK2 activity and membrane association at least under conditions of overexpression [12,15,16]. LRRK2 can phosphorylate Rab8a and Rab10, which may have downstream effects on the endolysosomal system. We, and others, have reported phosphorylation-dependent recruitment of Rab10 onto damaged lysosomes by LRRK2, in a process that orchestrates lysosomal homeostasis [17–19].

Rab8a is involved in a number of cellular functions including cell morphogenesis, neuronal differentiation, ciliogenesis, and membrane trafficking [20–22]. Rab8a regulates recycling of internalized receptors and membrane trafficking to the endocytic recycling compartment (ERC) [21,23]. Rab8a has been shown to interact directly with the transferrin receptor (TfR) and regulate polarized TfR recycling to cell protrusions [24] and transfer between cells through tunneling nanotubes [25]. Furthermore, Rab8a-depleted cells fail to deliver internalized TfR to the ERC [24]. Rab8a activity is controlled by its association with Rabin8, a guanine exchange factor [26]. Phosphorylation of Rab8a by LRRK2 inhibits its association with Rabin8 and thereby modulates Rab8a activity [8]. The G2019S LRRK2 mutation enhances Rab8a phosphorylation, mediating defects on EGFR recycling and endolysosomal transport [27]. Recently, mutations in LRRK2 were shown to mediate centrosomal deficits through Rab8a signaling [28–30], while ciliogenesis has been placed downstream of LRRK2 activity in different models [31].

Although these data indicate that Rab8a regulates exocytic and recycling membrane trafficking at the ERC, how phosphorylation of Rab8a affects downstream biology in PD-relevant cell types remains unclear. Here, we identify a novel role of LRRK2 in mediating Rab8a-dependent TfR recycling and show in vivo effects of LRRK2 mutation at the endogenous level on iron homeostasis in microglia. Expression of mutant LRRK2 induces sequestration of Rab8a to lysosomes and dysregulates transferrin recycling in a Rab8a-dependent manner. In induced pluripotent stem cell (iPSC)-derived microglia carrying the G2019S LRRK2 mutation, transferrin is mistrafficked to lysosomes in proinflammatory conditions, and this is accompanied

by subtle dysregulation of transferrin clearance compared to wild-type (WT) controls. Finally, we show that G2019S LRRK2 knock-in mice exhibit increased iron accumulation within microglia in the striatum in response to neuroinflammation compared to WT controls. These data support a role of LRRK2 in modulating normal iron uptake and storage in response to proinflammatory stimuli in microglia.

## Results

### Pathogenic mutant LRRK2 sequesters Rab8a to damaged lysosomes

In order to study endogenous Rab8a in cells, we first validated commercially available antibodies for western blotting and immunofluorescence employing small interfering RNA (siRNA)-mediated knock-down (S1 Fig). Using the validated antibody (Cell Signaling; D22D8; #6975), we examined the intracellular localization of Rab8a in the context of LRRK2 mutations. Endogenous Rab8a plays a part in receptor recycling and localizes in tubular recycling endosomes [21]. Consistent with such a function, in cells expressing WT LRRK2, Rab8a localized predominantly in tubular membranes not associated with Lamp2-positive lysosomes (Fig 1A). However, expression of R1441C or G2019S LRRK2 variants induced relocalization of endogenous Rab8a into enlarged perinuclear lysosomes (Fig 1A). Other pathogenic mutations, but not the kinase-dead variant K1906M, also induced relocalization of Rab8a (S2 Fig). The latter result suggested that the recruitment of Rab8a to lysosomes might be kinase dependent. To test this hypothesis, cells were treated with the LRRK2 kinase inhibitor MLi-2 for 1 hour prior to fixation and staining. LRRK2 kinase inhibition restored association of Rab8a with tubular recycling endosomes (Fig 1A). Our experiments show that around 60% of Rab8a staining colocalized with LRRK2 in cells expressing R1441C LRRK2 or G2019S LRRK2, and this was rescued back to WT LRRK2 levels (approximately 20%) by Mli-2 (Fig 1B). A parallel increase in colocalization was observed between Rab8a and Lamp2 in mutant LRRK2 expressing cells (Fig 1C). Recruitment of Rab8a by LRRK2 and colocalization with Lamp2 were rescued by MLi-2 treatment. Furthermore, we also quantified Rab8a recruitment by imaging Rab8a sequestration to large ($>4$ $\mu m^2$) cytoplasmic foci by high-content imaging (Cellomics, Thermo Fisher). We report a convergent phenotype for all PD-linked LRRK2 genetic variants with an increase in the fraction of cells that show a sequestered Rab8a phenotype while this was rescued by MLi-2 (S2B Fig). In our experiments, about 50% of cells expressing R1441C LRRK2 contained Rab8a-positive lysosomes, while this was true for about 30% of cells expressing G2019S LRRK2 and approximately 10% WT LRRK2 expressing cells (S2C Fig). To examine whether mutant LRRK2 and Rab8a are recruited to the lysosomal lumen or membrane, cells expressing G2019S LRRK2 were analyzed by superresolution microscopy (Airyscan), revealing a clear recruitment of both proteins to the membrane of Lamp1-positive lysosomes (Fig 1D).

Previous studies have reported recruitment of Rab8a to the perinuclear region in close proximity to centrosomes [29,30]; it is known that lysosomes are found in distinct perinuclear and peripheral pools and that the movement of lysosomes depends on microtubule-dependent transport and that the proportion of these 2 pools may differ between cancerous and nontransformed cells [32]. Furthermore, it is likely that permeabilization conditions used in fixed cell staining could cause rapid extraction of prenylated Rab GTPases from membranes that might bias which pool of Rab8a was readily imaged. To dissect lysosomal versus centrosomal pools of Rab8a in nontransformed cells, we compared standard immunocytochemistry protocols to live imaging of Rab8a in mouse primary astrocytes. G2019S LRRK2 was transiently expressed in primary mouse astrocytes, and endogenous Rab8a was stained along with Lamp1 (Fig 1E, left panel). For live imaging, transiently expressing a HaloTag pDEST G2019S LRRK2, visualized by JFX650 ligand, along with GFP-Rab8a and LAMP1-RFP were expressed and imaged

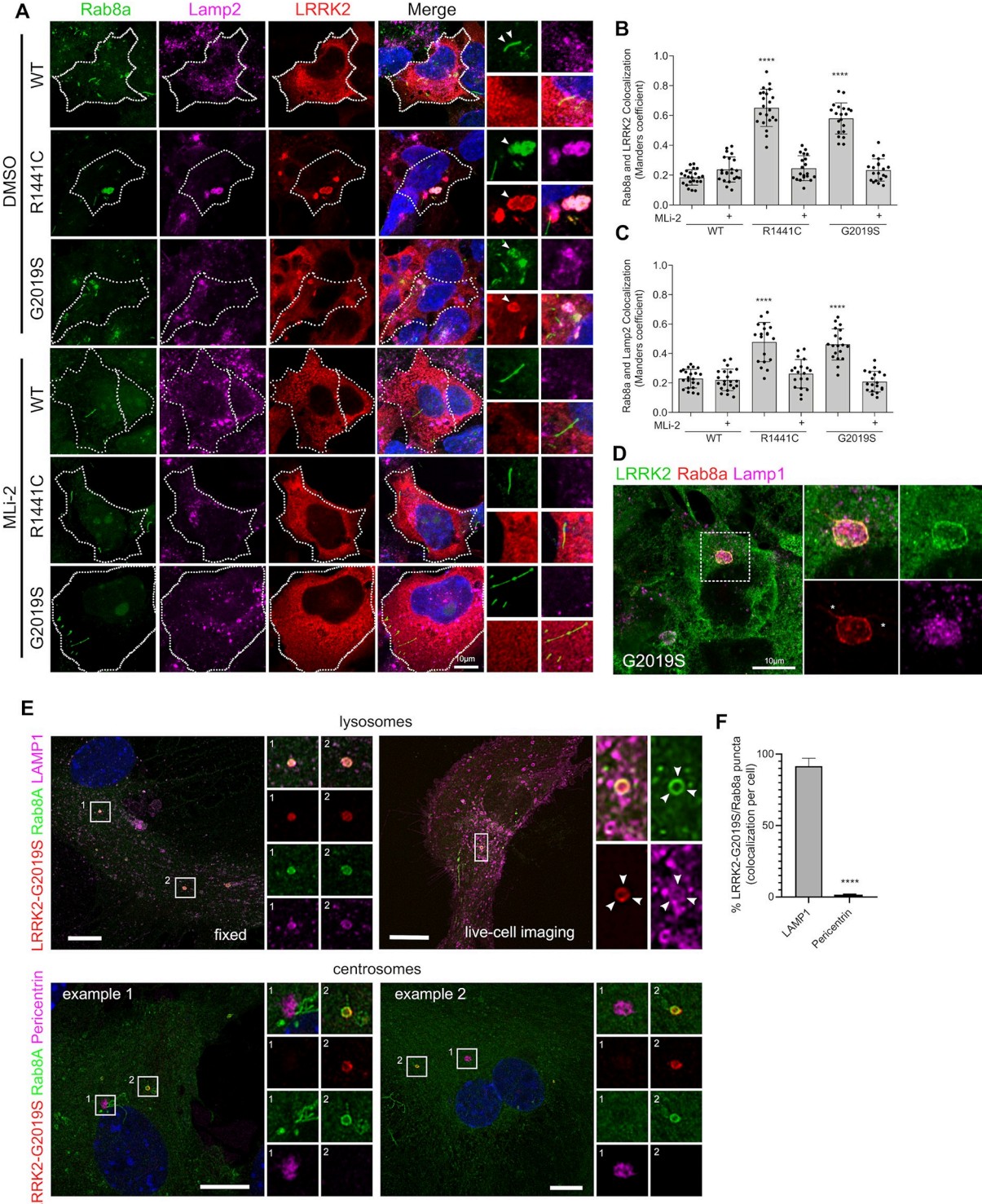

**Fig 1. Pathogenic mutations of LRRK2 sequester endogenous Rab8a to lysosomes in a kinase-dependent manner.** (A) Representative confocal images of HEK293T cells transiently expressing FLAG-tagged WT, R1441C, or G2019S LRRK2 constructs, stained for FLAG LRRK2, endogenous Rab8a, and endogenous Lamp2. Cells were treated with 1 μM MLi-2 for 1 hour or DMSO prior to staining, 24 hours posttransfection. (B, C) Quantification of Manders colocalization coefficient between Rab8a, LRRK2, and Lamp2 (B, C: $N > 20$ cells for each group across 2 independent experiments, $****P < 0.0001$, one-way ANOVA with Tukey post hoc test; B: $F(5, 120) = 108.9$; C: $F(5, 120) = 35.85$). (D) Superresolution confocal image of endogenous Rab8a and overexpressed G2019S LRRK2 localizing at the lysosomal membrane. (E) Mouse primary astrocytes were transfected with HaloTag-LRRK2(G2019S), GFP-Rab8a, and LAMP1-RFP (top panels). Cells were incubated

with JFX650 (100 nM) for 1 hour, washed, and imaged using a Nikon SoRa spinning disk microscope utilizing 3D Landweber deconvolution, 48 hours later. To analyze Rab8a localization to centrosomes, fixed cells were stained for pericentrin, following transient expression of G2019S LRRK2 and GFP-Rab8a. Images were taken with Airyscan (bottom panels). (F) Quantitation of the percentage of LRRK2/Rab8a puncta per cell that colocalizes with either LAMP1 or Pericentrin ($N > 19$ cells for each condition from 2 independent experiments; unpaired $t$ test; $P < 0.0001$). The underlying data can be found in S1 Data. LRRK2, leucine-rich repeat kinase 2; WT, wild-type.

(Fig 1E, right panel). In these experiments, fixed and live Rab8a imaging identically showed sequestration of Rab8a to the lysosomal membrane. In these nontransformed cells transiently expressing G2019S LRRK2, we did not detect Rab8a staining at the centrosome using pericentrin as a marker (Fig 1E and 1F). However, in HEK293T cells expressing G2019S LRRK2, we detected recruitment of endogenous Rab8a to the perinuclear region in close proximity to centrosomes (pericentrin staining), and this was partially rescued by nocodazole treatment (S3A and S3B Fig). Additionally, we found that approximately 50% of total Rab8a colocalizes with Lamp2 in the presence of G2019S LRRK2, and this is decreased to approximately 30% after treatment with nocodazole (S3C and S3D Fig). These data are in accordance with previous studies [17,30] and suggest that there are likely at least 2 pools of Rab8a in the presence of mutant LRRK2, one bound to the centrosome, and the other to lysosome-related structures.

Previous studies have shown that LRRK2 can localize in enlarged lysosomes along with Rab GTPases [17]. Furthermore, we have shown that LRRK2 can be recruited to damaged lysosomes that have low degradative capacity [18]. To further extend our previous findings in the context of Rab8a recruitment by LRRK2, we evaluated whether LRRK2-positive lysosomes were deficient in Cathepsin D. While LRRK2 mutations induced an increase in colocalization with Lamp2, minimal colocalization with Cathepsin D was detected in WT or pathogenic LRRK2 (Fig 2B), a finding also supported by superresolution imaging (Fig 2C). This result suggested that Rab8a was recruited specifically to damaged lysosomes. To further test this hypothesis, we treated cells with the lysosomal destabilizing agent LLOMe that interacts with the lysosomal membrane and luminal hydrolases. We found that LLOMe induced recruitment of WT LRRK2 to the membrane of enlarged lysosomes as previously shown [18] (Fig 2D). Endogenous Rab8a was also recruited to the lysosomal membrane in WT LRRK2 cells after treatment with LLOMe (Fig 2E). Furthermore, the cytosolic scaffolding protein JIP4 was also recruited to the same structure suggesting mistrafficking of factors involved in vesicle-mediated transport (Fig 2F). These data suggest that Rab8a is recruited to damaged lysosomes by mutant LRRK2 in a kinase-dependent manner.

## LRRK2-mediated T72 Rab8a phosphorylation blocks interaction with Rabin8 but not MICAL-L1

The above data suggest that Rab8a can move from its normal location at the ERC to the lysosome after expression of LRRK2 mutations. Given that all LRRK2 mutations are proposed to increase Rab phosphorylation, at T72 for Rab8a [8], we reasoned that the mechanism of relocalization might be related to this phosphorylation event. The pT72 Rab8a antibody that was available during this study was not specific to Rab8a and cross-reacted with Rab3A, Rab10, Rab35, and Rab43, which have a high degree of sequence conservation (for datasheet, refer to ab230260; Abcam). Therefore, using a tagged version of Rab8a, we found that exogenous expression of R1441C, Y1699C, G2019S, or I2020T LRRK2 induced a significant increase in Rab8a phosphorylation at T72 compared to WT or the kinase-dead variant K1906M, and this was blocked by MLi-2 treatment (Fig 3A and 3B).

Having established that Rab8a phosphorylation at T72 is increased by all pathogenic LRRK2 mutations, we next investigated how this affects interactions with known Rab8a

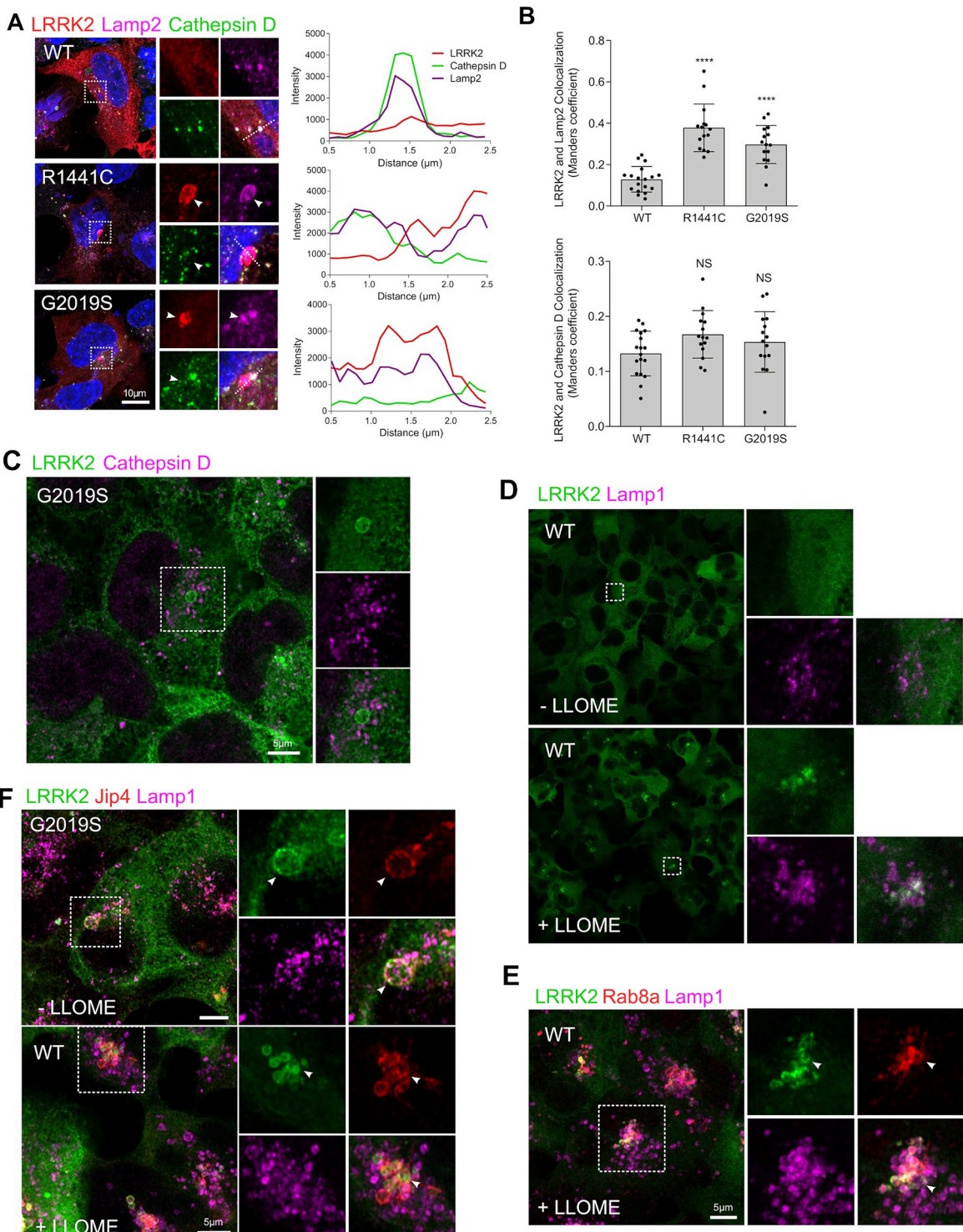

**Fig 2. LRRK2 and Rab8a are recruited to the membrane of damaged lysosomes.** (A) HEK293T cells transiently expressing FLAG-tagged WT, R1441C, or G2019S LRRK2 for 24 hours were stained for FLAG LRRK2, Lamp2, and Cathepsin D and analyzed by confocal microscopy. Staining intensity profiles were generated on sections indicated by the dotted lines. (B, C) Manders colocalization coefficient of LRRK2 versus Lamp2 or Cathepsin D staining (*N* > 20 cells per group across 2 independent experiments, ****P < 0.0001, one-way ANOVA with Tukey post hoc; Lamp2: F (5, 57) = 41.73; Cathepsin D: *NS*). (C) Superresolution image of HEK293T cells stably expressing GFP G2019S LRRK2 costained for Cathepsin D. HEK293T cells stably expressing GFP WT LRRK2 were treated with 1 mM LLOMe for 4 hours and stained for endogenous Lamp1 (D) and Rab8a (E). (F) GFP WT and G2019S expressing cells were treated with LLOMe as in

(E) and stained for Jip4 and Lamp1 prior to imaging by superresolution microscopy. The underlying data can be found in S1 Data. LRRK2, leucine-rich repeat kinase 2; WT, wild-type.

partners. We focused on the known activating GEF, Rabin8, and MICAL-L1, which is an established Rab8a-interacting partner [23] that localizes in tubular endosomes and is essential for efficient endocytic trafficking from the ERC to the plasma membrane [21]. Modeling the interaction between Rab8a and Rabin8, we predicted that the Threonine at position 72 on Rab8a resides 2.8 to 3.9 Å away from the neighboring Glutamate (E192) on Rabin8 (Fig 3C). Addition of a phosphate group on T72 will likely hinder the interaction with the negatively charged Glutamate, therefore destabilizing Rab8a–Rabin8 interaction. This prediction is consistent with work by Steger and colleagues who demonstrated biochemically that phosphorylation of Rab8a by LRRK2 can limit its activation by Rabin8 [8]. Threonine 72 of Rab8a is closest to a Phenylalanine (F647) on MICAL-L1 at a 6.7 to 8.4 Å distance, suggesting that phosphorylation at this site will not affect this interaction. To validate this prediction biochemically, we performed co-immunoprecipitation experiments by coexpressing Rab8a with WT, kinase-hyperactive I2020T, and kinase-dead K1906M LRRK2 in cells. In these conditions Rab8a was T72 hyperphosphorylated by I2020T LRRK2, and this modification did not hinder association with endogenous MICAL-L1 (Fig 3D). To investigate the retention of this association in the context of Rab8a recruitment to lysosomes, we stained for Rab8a and MICAL-L1 in cells expressing LRRK2. Cells expressing WT LRRK2 showed colocalization of Rab8a with MICAL-L1 in tubular membranes, while I2020T LRRK2 expression resulted in the recruitment of both proteins to LRRK2-positive structures consistent with the lysosomal phenotype described above (Fig 3E). This was also recapitulated by R1441C and G2019S LRRK2 that recruited endogenous MICAL-L1 and induced association of this trafficking protein with lysosomes (S4 Fig). Our data suggest that mutant LRRK2 can phosphorylate Rab8a inducing recruitment of both Rab8a and MICAL-L1 away from the ERC.

## Mutant LRRK2 sequesters Rab8a leading to transferrin mistrafficking and accumulation of intracellular iron

Given the above data showing that LRRK2 phosphorylation prevents binding of Rab8a to its effectors and redirects Rab8a and MICAL-L1 away from the ERC, we speculated that these events would then lead to a defect in Rab8a-mediated recycling. To evaluate this hypothesis, we examined the recycling of the transferrin receptor 1 (TfR), which is known to depend on Rab8a function [24]. Cells expressing LRRK2 genetic variants were stained for endogenous TfR and analyzed. TfR localized in distinct cytoplasmic vesicles in cells expressing WT LRRK2, whereas LRRK2 pathogenic mutants showed a clustered TfR localization that associated with LRRK2. The distribution of TfR vesicles in LRRK2-expressing cells was analyzed in Imaris (Bitplane, Zürich, Switzerland) where vesicles were rendered to spots throughout z-stack 3D reconstructed images and distances between them were measured (Fig 4A, right-hand panels). TfR vesicles were significantly more clustered in cells expressing LRRK2 mutants compared to WT LRRK2 expressing cells (Fig 4B and 4C). To test whether the coalescent TfR staining that we observe with mutant LRRK2 associates with lysosomes, we analyzed the fraction of TfR colocalizing with the late endosome/lysosome marker Lamp2 (S5 Fig). We found a significant increase in colocalization between TfR and Lamp2 in cells expressing R1441C (approximately 50% of whole-cell TfR staining) and G2019S LRRK2 (approximately 40%) compared to WT LRRK2 expressing cells (S5A and S3B Fig). Our data are consistent with prior reports of perinuclear localization of TfR in cells expressing kinase hyperactive mutant LRRK2 [31]. Our

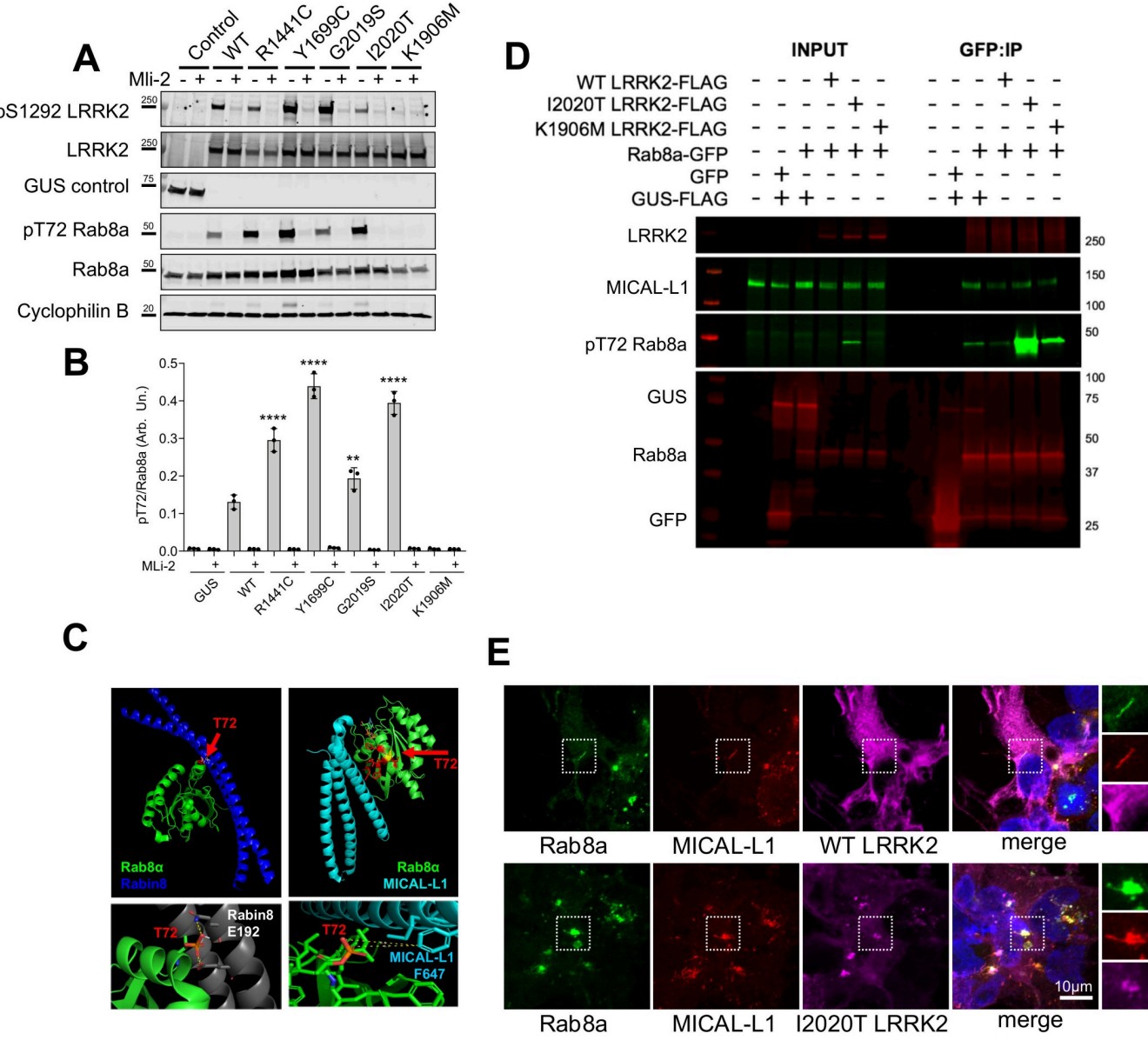

**Fig 3. Rab8a phosphorylation retains interaction with MICAL-L1 that is corecruited in mutant LRRK2 expressing cells.** (A) HEK293T cells expressing control (GUS), FLAG WT, or mutant LRRK2 variants were treated with MLi-2 prior to lysis and analysis by western blotting for pS1292 and total LRRK2, as well as pT72 and total Rab8a. (B) Quantification of pT72 Rab8a levels normalized to total Rab8a (B, two-way ANOVA; $N = 3$ independent experiments; treatment: $P < 0.0001$, F (1, 28) = 1473, genotype: $P < 0.0001$, F (6, 28) = 157.1). (C) Structural modeling of T72 phosphorylation on Rab8a in association with Rabin8 or MICAL-L1. T72 is between 2.8–9.9 Å from the closest glutamate on Rabin8 and 6.7–8.4 Å from the closest phenylalanine on MICAL-L1. (D) HEK293T cells expressing FLAG WT, I2020T, or K1906M LRRK2 along with GFP Rab8a, GFP control, or FLAG-GUS control were lysed and proteins immunoprecipitated using GFP beads. Co-immunoprecipitated proteins were analyzed by western blotting probed for MICAL-L1 as well as pT72 and total Rab8a. (E) Cells expressing FLAG WT or I2020T LRRK2 were stained for endogenous Rab8a and MICAL-L1 and analyzed by confocal microscopy. The underlying data can be found in S1 Data. LRRK2, leucine-rich repeat kinase 2; WT, wild-type.

data suggest that TfR is recruited to damaged lysosomes by mutant LRRK2. To further test this hypothesis, we treated cells with LLOMe and analyzed intracellular localization by Airyscan (S5C Fig). We observe TfR recruitment in LRRK2-positive and Rab8a-positive membranes consistent with damaged lysosomes as previously described [18].

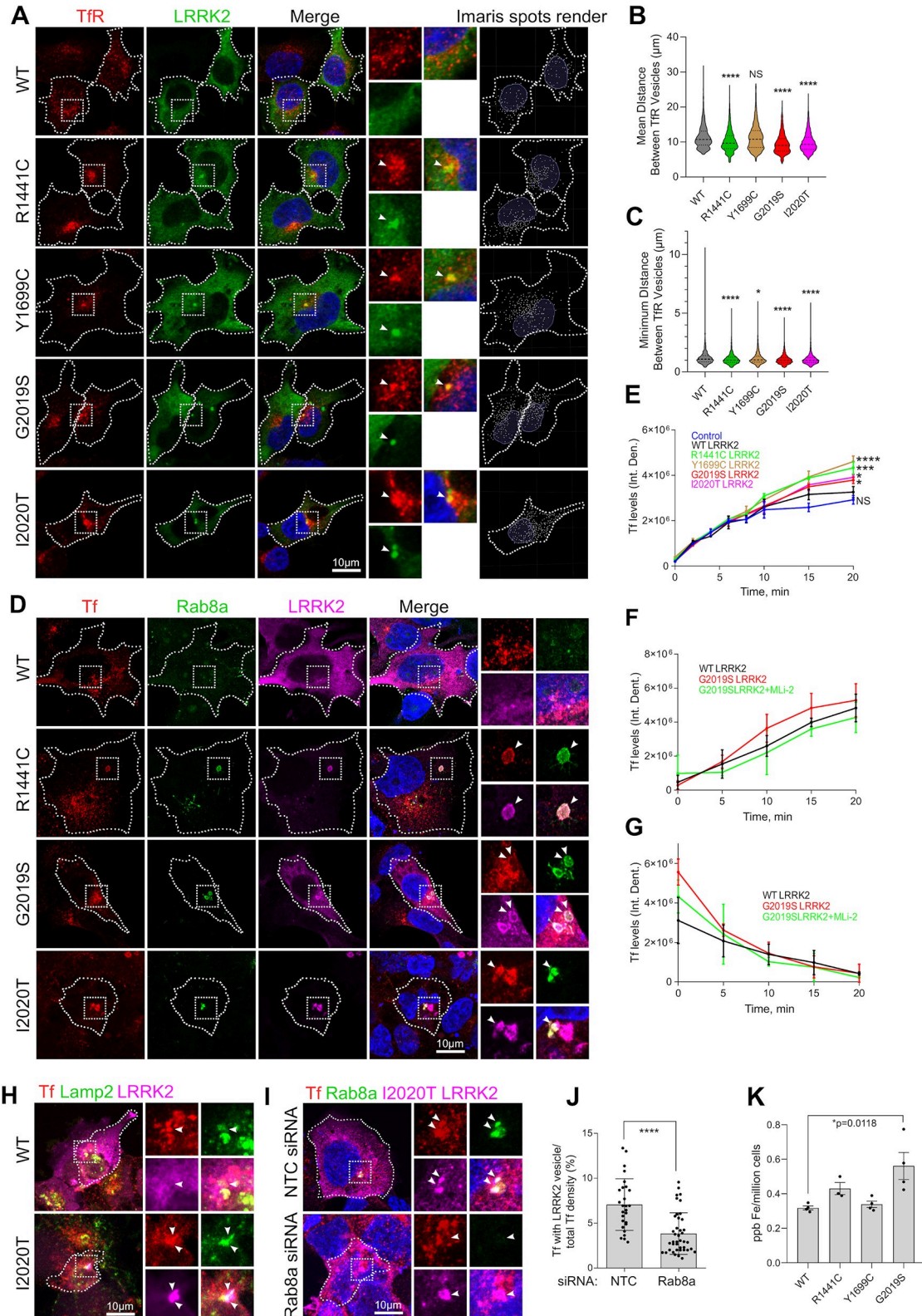

**Fig 4. Mutant LRRK2 sequesters TfR and dysregulates transferrin recycling.** (A) HEK293T cells exogenously expressing WT and mutant LRRK2 constructs were stained for TfR and visualized by superresolution microscopy. TfR vesicles were analyzed using the Imaris Spot Detection module, and the mean and minimum distances between spots were plotted (B, C). ($N > 2,000$ vesicles were

counted in at least 20 cells per construct from 2 independent experiments, $^*P < 0.05$, $^{****}P < 0.0001$, one-way ANOVA with Tukey post hoc, B: F(4, 13334) = 46.14, C: F (4, 13336) = 194.1). (D) HEK293FT cells expressing LRRK2 genetic variants were incubated with Alexa Fluor 568–conjugated transferrin, fixed at 20 minutes of incubation and stained for endogenous Rab8a and FLAG LRRK2 (D). (E) Uptake of Alexa Fluor 568–conjugated transferrin was monitored by high-content imaging, and transferrin levels per cell were plotted at different time points (E: T = 20 minutes, $N$ = 3 technical replicates per construct (>800 cells/well) one-way ANOVA, Tukey post hoc, $^*P < 0.05$, $^{***}P = 0.0005$, $^{****}P < 0.0001$). (F) HEK293T cells exogenously expressing FLAG WT or G2019S LRRK2 constructs were incubated in DMEM supplemented with 1 μM MLi-2 or DMSO for 45 minutes prior to addition of Alexa Fluor 568–conjugated transferrin in the same media, and high-content imaging was used to monitor Tf uptake at different time points (F: $N$ = 3 technical replicates per construct (>800 cells/well), two-way ANOVA, genotype: $P < 0.05$, F (2,30) = 3.575; Time: $P < 0.0001$, F (4, 30) = 44.96). (G) Cells were treated with MLi-2 as in (F), and incubated with Alexa Fluor–conjugated transferrin for 30 minutes, prior to changing to fresh media containing MLi-2 and monitoring transferrin release by high-content imaging (G; at T10: $N$ = 3 technical replicates per construct (>800 cells/well), one-way ANOVA, Tukey post hoc, $P < 0.05$, F (2, 30) = 3.575). (H) HEK293T cells transiently expressing WT or I2020T LRRK2 were stained for Tf, Lamp2, and LRRK2 (FLAG) and analyzed by confocal microscopy. (I, J) Cells were transfected with Rab8a siRNA or scrambled sequence constructs (control) and 24 hours later were transfected with I2020T LRRK2 that was expressed overnight, before fixation and staining for Tf, Rab8a, and LRRK2 (FLAG). ($N$ = 27 cells for NTC siRNA, $N$ = 41 cells for Rab8a siRNA imaged across 2 independent experiments, two-tailed Mann–Whitney $U$ test, $^{****}P < 0.0001$). Intracellular iron levels were analyzed by ICP-MS in cells stably expressing GFP WT or mutant LRRK2 constructs (K) ($N$ = 4 confluent plates of cells per construct, two-way ANOVA with Tukey post hoc, $^*P = 0.018$, F (3, 11) = 6.225). [SD bars are shown]. The underlying data can be found in S1 Data. ICP-MS, inductively coupled plasma mass spectrometry; LRRK2, leucine-rich repeat kinase 2; siRNA, small interfering RNA; TfR, transferrin receptor; WT, wild-type.

The observed altered TfR intracellular distribution in cells transiently expressing mutant LRRK2 prompted us to test whether LRRK2 functionally affected Rab8a-dependent Tf recycling. Cells expressing WT or mutant LRRK2 were preloaded with Alexa Fluor 568-Tf and visualized at T10 minutes of recycling in fresh media. Mutant LRRK2 expressing cells exhibited association of Tf with enlarged vesicles that were labeled with LRRK2 and Rab8a (Fig 4D). Tf vesicles were clustered in G2019S LRRK2 expressing cells compared to WT LRRK2, while this was rescued by treatment with MLi-2 prior to Tf loading (S5D Fig). In Tf uptake time course experiments, LRRK2 mutant expressing cells plateaued at higher Tf levels compared to WT at T15 to T20 minutes as monitored by high-throughput imaging, pointing to Tf accumulation driven by dysregulated recycling (Fig 4E). The modest increase in net transferrin levels detected in G2019S LRRK2 expressing cells was partly reversed in cells treated with MLi-2 (Fig 4F). We further evaluated a possible impairment of Tf recycling employing pulse-chase experiments to monitor Tf clearance. Tf recycling assays revealed higher initial transferrin levels at T0 following 45 minutes of loading in G2019S LRRK2 expressing cells compared to WT, while this difference was not evident at later time points (Fig 4G). In parallel experiments, we detected no significant difference in the total levels or surface-bound levels of TfR in cells expressing G2019S LRRK2 and WT LRRK2 (S5E Fig). These data cumulatively suggest a subtle dysregulation of transferrin clearance driven by transient expression of mutant LRRK2.

The intracellular localization of internalized Tf in the context of LRRK2 mutations was explored further. In cells overexpressing I2020T LRRK2, we observed partial colocalization of internalized Tf with Lamp2 (Fig 4H), suggesting that excess Tf may associate with lysosomes. To test whether this phenotype was Rab8a dependent, the localization of internalized Tf in I2020T LRRK2 expressing cells was investigated following Rab8a siRNA knock-down. Knock-down of Rab8a decreased the proportion of Tf that colocalized with mutant LRRK2 relative to total internalized Tf (Fig 4I and 4J). Lastly, inductively coupled plasma mass spectrometry (ICP-MS) analysis revealed elevated iron levels in cells stably expressing G2019S LRRK2 compared to WT LRRK2, while the pathogenic R1441C and Y1699C mutants do not show detectable alterations in iron levels, at least as measured by ICP-MS (Fig 4K). Collectively, these data suggest subtle dysregulation of Tf-mediated iron uptake and iron homeostasis by LRRK2 mutations as a consequence of altered Rab8a localization away from the ERC and toward lysosomes.

## Endolysosomal and iron-binding gene expression in microglia is modulated by inflammation, in vitro and in vivo

LRRK2 is expressed in microglia and has been linked to inflammation, cytokine release, and phagocytosis. Our data presented here show that LRRK2 associates with lysosomes and that LRRK2 mutations dysregulate Tf-dependent iron uptake mechanisms. Glial cells are known to be iron rich, and microglia activation state is integrally linked to brain iron content [33]. Proinflammatory stimuli are known to induce uptake of extracellular iron by microglia, and, conversely, the iron status of their environment can modulate their activation. Given the role of LRRK2 in inflammation, we asked whether endolysosomal processes and iron homeostasis might converge in proinflammatory conditions. We addressed this hypothesis using 2 genome-scale approaches. First, we mined our previously described RNA-Seq dataset [34] of primary microglia treated with lipopolysaccharide (LPS) or preformed α-synuclein fibrils in culture (Fig 5A). Analysis of the shared hits between the 2 treatments using gene ontology revealed enrichment for endosomal and lysosomal pathways (Fig 5B and 5C). Unsupervised hierarchical clustering of differential gene expression separated the controls from the 2 treatment groups (Fig 5D). These data demonstrate that genes involved in the endolysosomal system are consistently regulated by inflammatory stimuli.

We next asked whether the same regulation could be confirmed in vivo. Either WT or Lrrk2 knockout (KO) mice were given a single intrastriatal injection of LPS or vehicle control and 3 days later adult microglia were isolated using CD11b microbeads (Fig 5E). Between 3,000 and 5,000 cells were recovered from each of the 4 groups and analyzed using single-cell RNA-Seq (scRNA-Seq). We identified 5 distinct clusters of cells based on transcriptome similarity that represent resting microglia and 4 activation states (Fig 5F), with PBS-injected animals exhibiting predominantly resting microglia whereas the LPS-injected animals spanned the range of activation states (Fig 5G). Notably, we did not see strong differences between genotypes other than the KO animals tended to have fewer activated microglia, suggesting that Lrrk2 influences how microglia respond to proinflammatory stimuli [35]. Gene ontology showed enrichment for lysosomal and endocytic membrane processes (Fig 5H). These analyses identified genes involved in iron storage and uptake (FTH1; ferritin heavy chain), vesicular transport (SNAP23), acidification of vesicles (ATP6V1G1), and the late endosomal pathway (Rab7) (Fig 5I). These data demonstrate that processes of vesicular trafficking and iron homeostasis are modulated in a synchronized manner by inflammation. Given the importance of iron uptake in microglia activation, we hypothesized that LRRK2 may play a role in trafficking of TfR and that PD-linked mutations may affect iron uptake and accumulation in inflammatory conditions. Thus, we next sought to examine the effect of LRRK2 mutations on Tf recycling and iron uptake in proinflammatory conditions.

## G2019S LRRK2 induces transferrin mislocalization and association with lysosomes in iPSC-derived microglia

To examine whether endogenous LRRK2 mutations might influence Tf-mediated iron uptake as modulated by neuroinflammation, iPSC-derived microglia from G2019S carriers were examined in resting and proinflammatory conditions, and Tf localization was assessed by superresolution microscopy (characterization of iPSC-derived microglia is shown in S6 Fig). Endogenous Tf showed partial colocalization with lysosomes in control conditions that was significantly decreased following LPS treatment, reflecting induction of iron uptake and recycling close to the plasma membrane, in WT cells. In contrast, the G2019S LRRK2 cells retained association of Tf with lysosomes following LPS treatment (Fig 6A and 6B). After 3D rendering (Fig 6A, right-hand panels; S1–S4 Movies for Imaris processing), Tf vesicle size and proximity

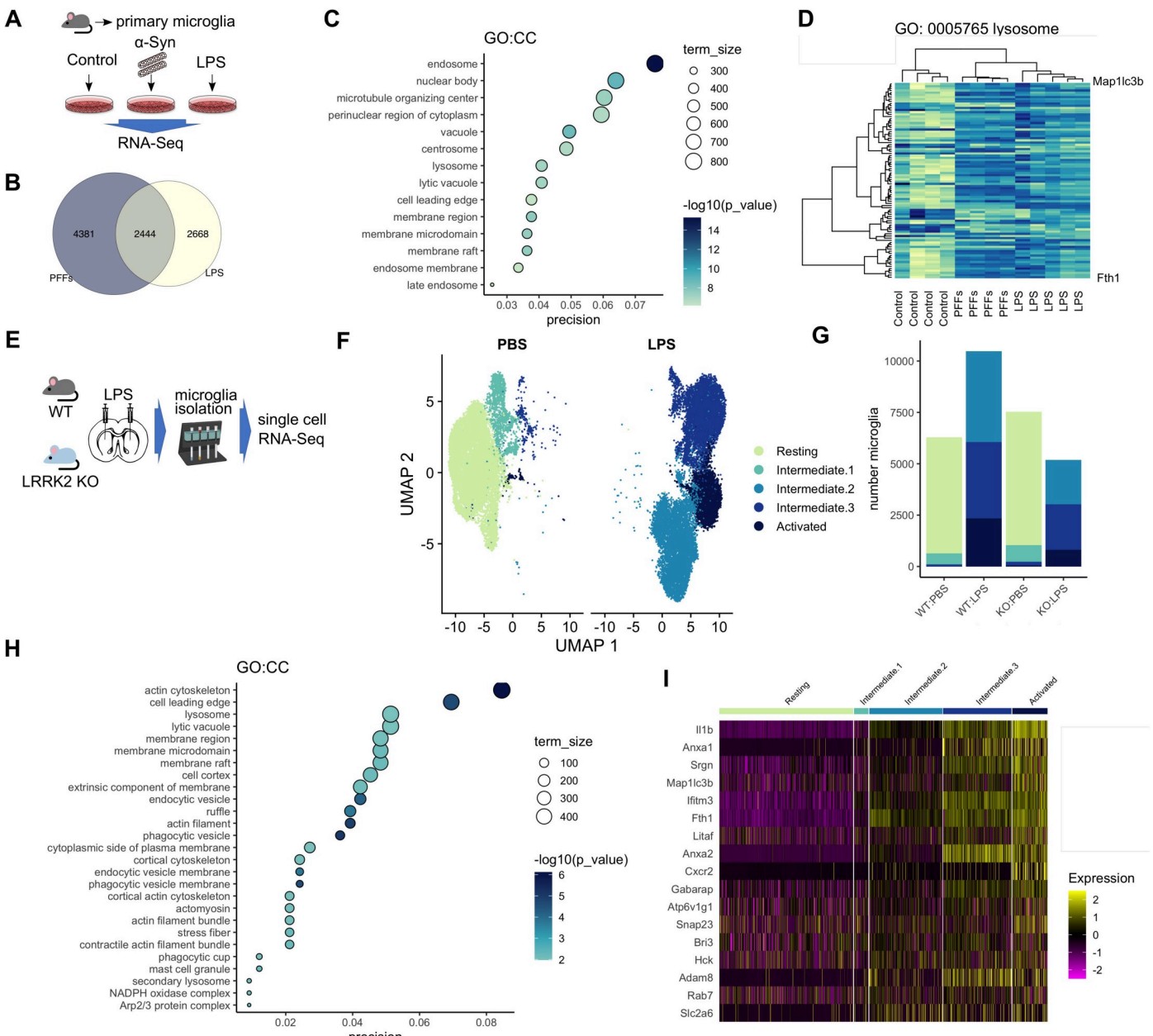

**Fig 5. Neuroinflammation remodels endolysosomal gene expression in microglia, in vitro and in vivo.** (A) Outline of RNA-Seq experiment: Primary mouse microglia cultures were incubated with LPS or α-synuclein fibrils, and transcriptomic profiles were analyzed by RNA-Seq. (B) Common and distinct hits were detected between the LPS and PFF-treated groups. (C) Bubble plot showing GO:CC term enrichment in the shared hits from LPS and PFF-treated primary microglia highlights enrichment for endolysosomal processes. (D) Unsupervised hierarchical clustering shows that the treated groups cluster together suggesting common transcriptomic profiles. (E) Schematic of the in vivo experiment where LPS striatal injections were administered to WT and LRRK2 KO mice, followed by microglia isolation and single-cell RNA-Seq. (F) UMAP plot showing separation of the retrieved microglia in distinct groups of activation states. (G) Microglia from LPS-injected animals spanned the activation states, while PBS-injected animals gave predominantly resting microglia. (H) Lysosomal and endocytic mechanisms as well as cytoskeletal pathways are enriched in the cumulative data. (I) Heatmap showing clustering of microglia in distinct activation states highlighting increase in lysosomal and iron-related gene expression by inflammation. The underlying data have been deposited in NCBI's GEO [75] and are accessible through GEO accession numbers GSE186483 and GSE186559. GEO, Gene Expression Omnibus; KO, knockout; LRRK2, leucine-rich repeat kinase 2; LPS, lipopolysaccharide; PFF, preformed fibril; scRNA-Seq, single-cell RNA-Seq; WT, wild-type.

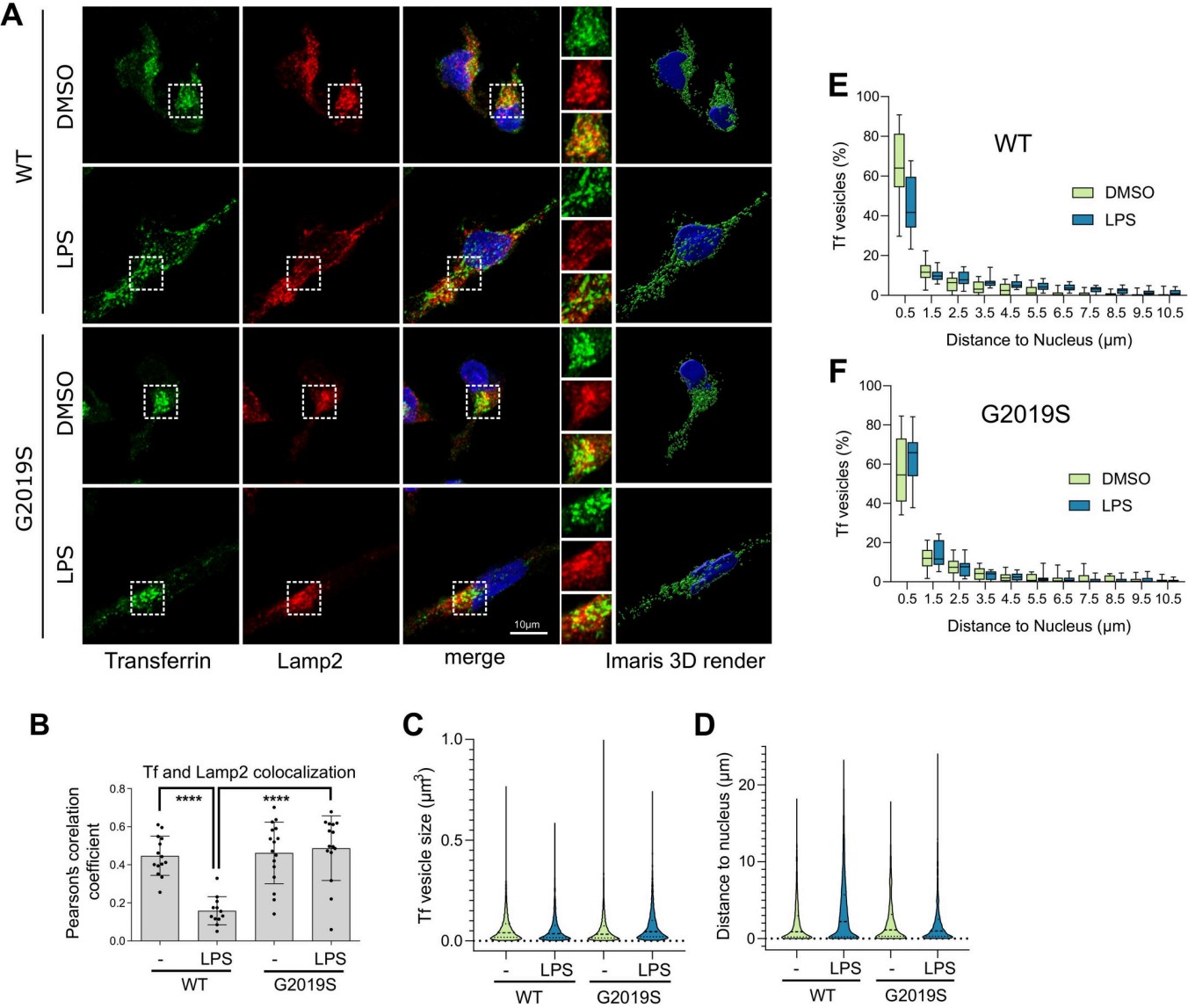

**Fig 6. G2019S LRRK2 modulates Tf recycling in iPSC-derived human microglia.** (A) iPSC-derived human microglia from WT or G2019S LRRK2 carriers were treated with LPS, and the localization of endogenous Tf and Lamp2 was analyzed by superresolution microscopy and the Imaris Surface render module (A). Partial colocalization between Tf and Lamp2 was observed in control that was significantly decreased with LPS treatment in WT cells but not in G2019S LRRK2 cells that retained lysosomal association of Tf (B) (N > 12 cells per group from 2 differentiations, one-way ANOVA Tukey post hoc, ****P < 0.0001, F (3,53) = 16.16). G2019S LRRK2 iPSC microglia exhibited larger Tf vesicles compared to WT while LPS treatment induced a decrease in average vesicle size in both cohorts (C) (minimum 3,000 vesicles were counted from 16 cells per group from 2 differentiations, two-way ANOVA, genotype: P < 0.0001, F(1,17753) = 16.38, treatment: P < 0.0001, F(1,17753) = 20.57). LPS treatment induced an increase in the average distance of Tf vesicles from the nucleus in WT cells but that was not significant in G2019S LRRK2 cells (D) (minimum 3,000 vesicles were counted from 16 cells per group from 2 differentiations, two-way ANOVA, genotype: P < 0.0001, F(1,14735) = 104.9, treatment: P < 0.0001, F (1, 14735) = 105.0). The frequency distributions of Tf vesicle proximity to the nucleus were plotted in E and F. The percentage of Tf vesicles proximal to the nucleus was significantly decreased with LPS treatment in WT but not in G2019S LRRK2 cells (E, F) (E: bin at 0.5 μm, two-tailed Student t test; Mann–Whitney U post hoc; **P = 0.0022; F: bin at 0.5 μm, two-tailed Student t test; Mann–Whitney U post hoc; NS). The underlying data can be found in S1 Data. iPSC, induced pluripotent stem cell; LPS, lipopolysaccharide; LRRK2, leucine-rich repeat kinase 2; Tf, transferrin; WT, wild-type.

to the nucleus were measured (Fig 6C and 6D). The G2019S LRRK2 cells showed significantly larger Tf vesicles that were sequestered closer to the nucleus compared to WT cells (Fig 6C and 6D). Additionally, in resting conditions, around 50% to 60% of Tf vesicles were concentrated near the juxtanuclear region, positioned within 1.5 μm from the nuclear membrane, in

both WT and G2019 LRRK2 cells. Upon activation by LPS, WT LRRK2 cells exhibited a more dispersed Tf localization throughout the cytoplasm with a lower fraction of vesicles close to the perinuclear recycling compartment (approximately 40% within 1.5 μm), while in contrast, G2019S LRRK2 cells showed an increase in proximity to the nucleus (Fig 5E and 5F). These data suggest that in G2019S LRRK2 cells, Tf was retained in the perinuclear recycling compartment region associated with lysosomes, under proinflammatory conditions. To characterize the dynamics of Tf trafficking in this model, we assayed Tf uptake and recycling by high-content imaging of cells in resting or proinflammatory conditions. Tf uptake levels per cell showed variability in our cultures, and no significant difference between conditions or genotype was observed (S7A and S7B Fig). While cell-to-cell variability was also noted in pulse-chase experiments, we detected a modest increase in Tf in G2019S LRRK2 cells under LPS treatment and a subtle effect of slower recycling compared to the other groups (S7C and S7D Fig). Our data support a model whereby endogenous LRRK2 mutations dysregulate recycling mechanisms and redirect Tf receptors to lysosomes under inflammatory conditions in microglia, predicting impairment of iron homeostasis. We next tested whether we could support this hypothesis in vivo using the G2019S LRRK2 knock-in mouse model that expresses this mutation in the appropriate endogenous context.

## G2019S LRRK2 induces iron and ferritin accumulation in inflammatory microglia in vivo

To examine the effect of LRRK2 mutations on iron accumulation in inflammation, we again used intrastriatal LPS injections on age-matched WT, Lrrk2 KO, and G2019S knock-in mice. Perls staining was used to visualize iron deposition in sections that spanned the striatum and substantia nigra (SN) 72 hours after injection (Fig 7A). The G2019S LRRK2 knock-in mice showed a marked increase in iron deposition in the striatum compared to WT and Lrrk2 KO mice as analyzed by densitometry (Fig 7B and 7C). Higher-magnification images revealed that the Perls stained cells had a microglial morphology, in line with our in vitro data that suggest dyshomeostasis of iron regulation mechanisms in inflammatory microglia (Fig 7D).

Additional sections from the same animals were stained for FTH1 and Tf. Accumulation of ferritin was observed in G2019S LRRK2 mice compared to WT and LRRK2 KO mice, with patterns similar to that seen with iron deposition (Fig 8A and 8B) while Tf was not significantly altered in the knock-in model (Fig 8C). To characterize further the type of cells that accumulate ferritin, sections were costained for Iba1 and analyzed by imaging (Fig 8D). Iba1-positive microglia were positive for ferritin in both WT and G2019S LRRK2 LPS-treated cohorts. About 50% of ferritin-positive cells were Iba1 positive in both groups (Fig 8E). Combined with the enhanced overall intensity of ferritin staining, we infer that these results are not due to increased numbers of Iba1-positive microglia but rather due to increased ferritin expression per cell. These data suggest that microglia, potentially among other cell types, accumulate ferritin in the striatum upon LPS-induced inflammation and drive ferritin-bound iron accumulation in G2019S LRRK2 knock-in mice compared to WT mice.

## Discussion

In this study, we show that PD-linked LRRK2 mutations have a convergent phenotype of Rab8a mislocalization away from the ERC and recruitment to damaged lysosomes that is distinct from WT protein. This relocalization is associated with dysregulation of Rab8a-mediated transferrin recycling both in heterologous cell lines and in human iPSC-derived microglia from PD patients with LRRK2 mutations following inflammatory stimulus (Fig 9 for model schematic). Furthermore, we show that in G2019S LRRK2 knock-in mice, LPS-induced

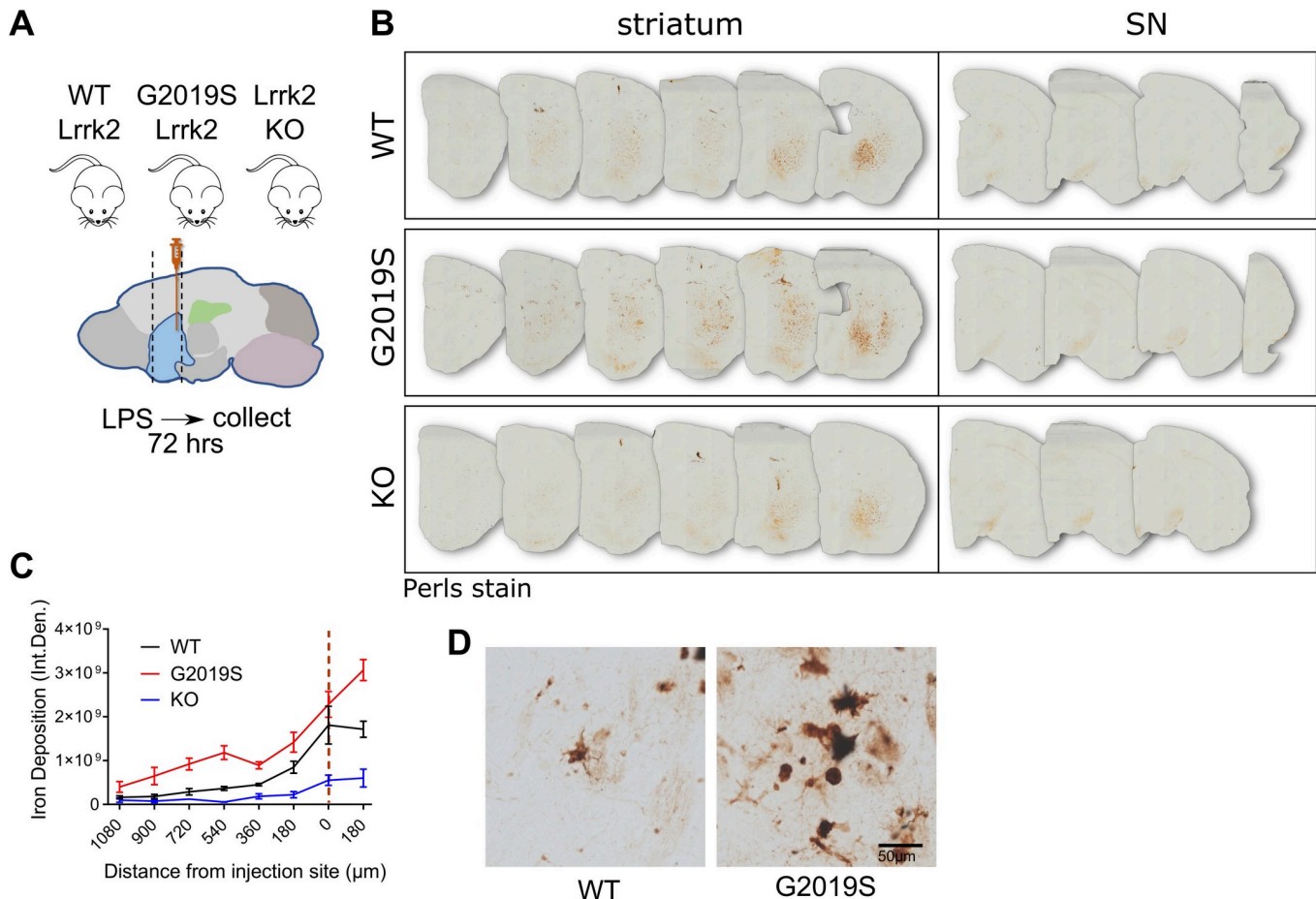

**Fig 7. G2019S LRRK2 induces iron accumulation in inflammatory microglia in vivo.** (A) Schematic of experimental design: WT, G2019S knock-in, and Lrrk2 KO mice were administered intrastriatal injections of LPS, and 72 hours later, brains were collected and stained by Perls stain. (B, C) G2019S LRRK2 knock-in mice exhibited significantly higher iron deposition in the striatum proximal to the injection site compared to WT and Lrrk2 KO, while minimal signal was detected in the SN in all groups ($N$ = 4 WT mice, 4 G2019S and 2 Lrrk2 KO; two-way ANOVA, Tukey post hoc, Genotype: ***$P$ = 0.0007, F (2, 7) = 24.70, Distance from injection site: ****$P$ < 0.0001, F (7, 49) = 32.28). High-magnification images revealed iron deposition in inflammatory microglia (D). [SEM bars are shown]. The underlying data can be found in S1 Data. KO, knockout; LPS, lipopolysaccharide; LRRK2, leucine-rich repeat kinase 2; SN, substantia nigra; WT, wild-type.

inflammation in the striatum results in higher iron accumulation in microglia and increased ferritin staining in vivo. These results suggest that LRRK2 plays a role in iron homeostasis response in neuroinflammation, driven, at least in part, by altered endolysosomal functions.

Transferrin-mediated uptake is the main route of iron delivery to most cell types. Extracellular transferrin binds ferric iron ($Fe^{3+}$) and is internalized by TfR through clathrin-mediated endocytosis. TfR is trafficked through early endosomes and can either be rapidly recycled back to the membrane or associate with the ERC in a slower recycling step. In the acidic lumen of endolysosomes, ferric iron ($Fe^{3+}$) is reduced to ferrous iron ($Fe^{2+}$), which mediates its release from Tf and the export to the cytosol through DMT1, where iron is subsequently delivered to different subcellular components. Lysosomal pH regulates iron release and impaired lysosomal acidification has been reported to trigger iron deficiency and inflammation in vivo as well as mitochondrial defects [36,37]. As LRRK2 mutations have been reported to alter the autophagic lysosomal pathway in carriers [38,39], as well as to perturb lysosomal acidification in knock-in mouse models [40], it is plausible that LRRK2 mutations affect Tf-mediated iron uptake by

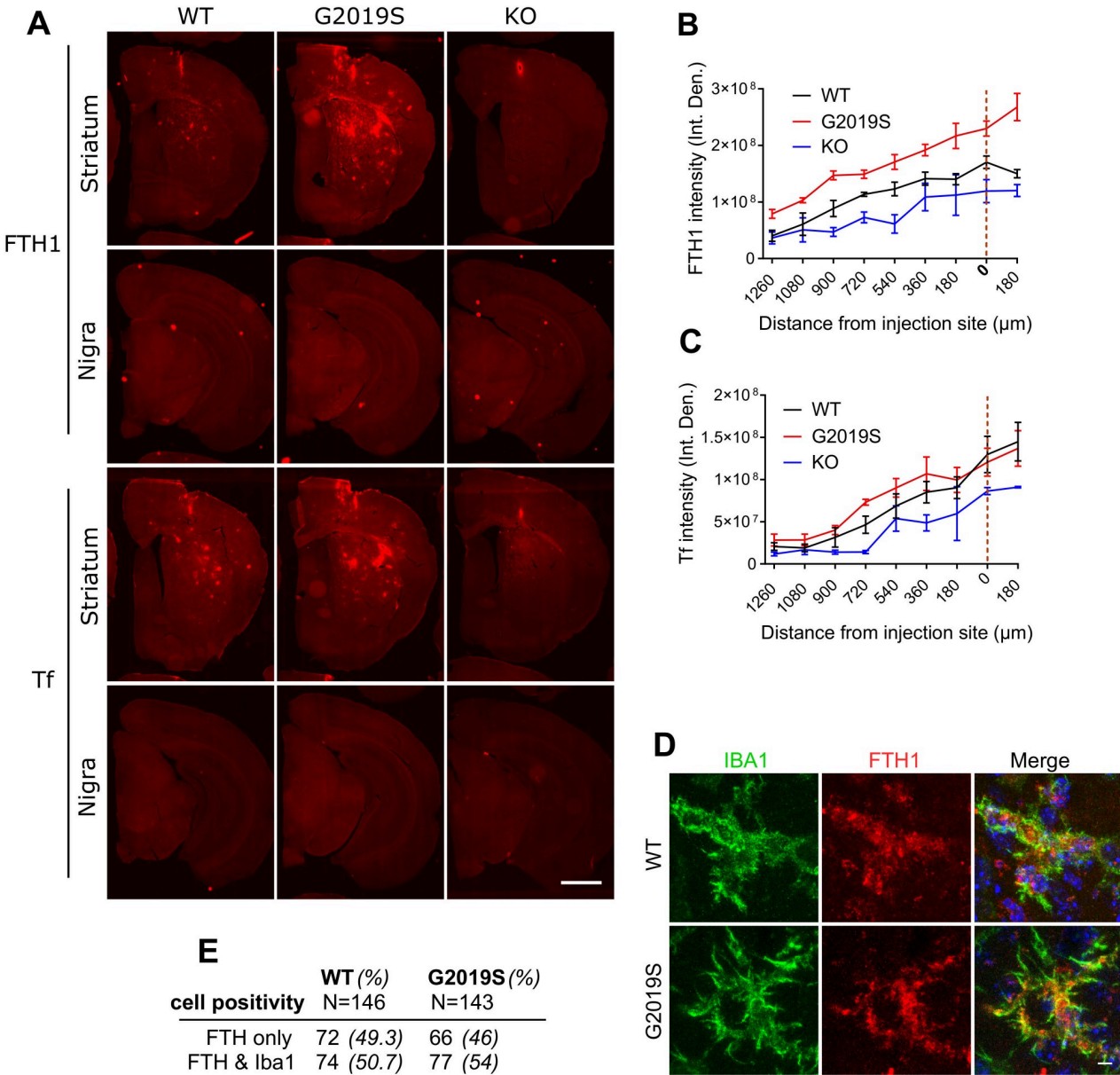

**Fig 8. Inflammation induces ferritin accumulation in microglia in G2019S LRRK2 knock-in mice.** FTH and Tf were stained and visualized in collected brains, 72 hours after post-intrastriatal injections of LPS. G2019S LRRK2 mice exhibited higher levels of FTH across the striatum compared to WT and Lrrk2 KO mice while Tf was not altered significantly in the knock-in (A, B, C) ($N$ = 4 WT mice, 4 G2019S and 2 Lrrk2 KO; B: two-way ANOVA, Tukey post hoc, Genotype: $**P$ = 0.0017, $F_{(2, 7)}$ = 18.27, Distance from injection site: $****P < 0.0001$, $F_{(8, 56)}$ = 50.25; C: two-way ANOVA, Tukey post hoc, Genotype: $P$ = 0.1040, $F_{(2, 7)}$ = 3.182, Distance from injection site: $****P < 0.0001$, $F_{(8, 56)}$ = 29.17). Microglia are positive for FTH in both WT and G2019S LRRK2 cohorts (D). Similar percentages of FTH-positive microglia were observed between the WT and G2019S LRRK2 groups (E). [SEM bars are shown]. The underlying data can be found in S1 Data. FTH, ferritin heavy chain; KO, knockout; LPS, lipopolysaccharide; LRRK2, leucine-rich repeat kinase 2; Tf, transferrin; WT, wild-type.

modulating lysosomal acidification. Furthermore, other studies have reported enlarged and damaged lysosomes in the context of LRRK2 mutations, consistent with our data [41,42]. In our experiments, we observe sequestration of TfR and accumulation to lysosomes driven by mutant LRRK2, while no significant difference was found on total or membrane-bound TfR levels in these cells or in G2019S knock-in mice that exhibited iron deposition and ferritin up-

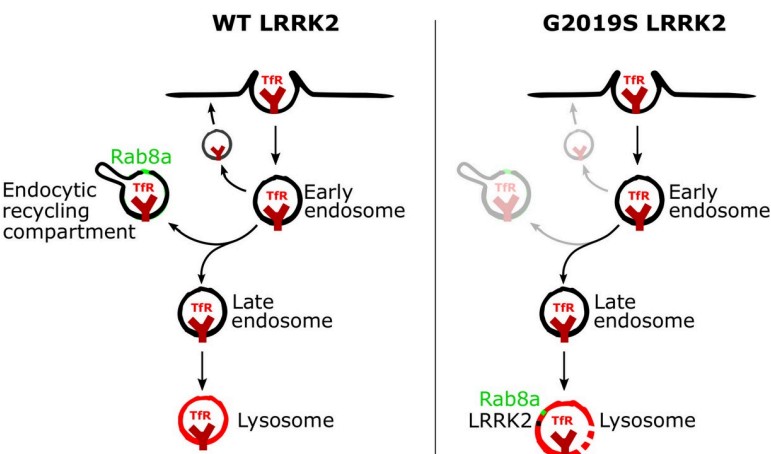

**Fig 9. A model of the recruitment of Rab8a to lysosomes and dysregulation of transferrin recycling by mutant LRRK2.** Following clathrin-mediated endocytosis, TfR can undergo (a) rapid recycling close to the membrane; (b) slower recycling via the ERC; or (c) targeting to lysosomes. Rab8a mediates recycling of TfR via the ERC, and hyperphosphorylation of Rab8a by mutant LRRK2 induces sequestration to damaged lysosomes and dysregulation of Rab8a-mediated trafficking. ERC, endocytic recycling compartment; LRRK2, leucine-rich repeat kinase 2; TfR, transferrin receptor; WT, wild-type.

regulation following LPS administration. Consistent with these observations, TfR levels are not being found to be increased in PD or in mouse models of the disease [43,44], suggesting that the regulation of transferrin recycling through the endolysosomal system rather than overall levels contribute to iron dyshomeostasis. While our data support the notion that pathogenic LRRK2 causes sequestration of TfR and accumulation in degradation-deficient lysosomes, it remains possible that the identity of this vesicular compartment is not lysosomal but endosomal in nature, as a significant portion of Lamp1 can reportedly colocalize with early endosomes in neurons [45]. LRRK2 deficiency also reportedly impairs recycling of TfR and is required to maintain the lysosomal degradative capacity of endocytic and autophagic cargo [46]. The LRRK2 G2019S mutation can affect Rab8a-mediated receptor recycling and endolysosomal transport [29]. We have previously reported that LRRK2 mutations can impair clathrin-mediated endocytosis in cell models, which governs internalization of different receptors including TfR [47].

We have recently shown that chronic pharmacological LRRK2 inhibition in G2019S Lrrk2 knock-in mice induces dysregulation of endolysosomal processes and subtle changes in mitochondrial homeostasis factors, further establishing a link between LRRK2 kinase activity and trafficking pathways that can control iron homeostasis [48]. Our data support an effect of LRRK2 mutations in dysregulating trafficking of internalized transferrin, promoting association with lysosomes. Additionally, we observe concomitant up-regulation of net intracellular iron in cells expressing G2019S LRRK2. It is important to note that in the ICP-MS experiment, we extracted and assayed whole-cell iron and thus cannot infer which cellular organelles are responsible for the observed up-regulation in G2019S LRRK2 expressing cells. It is known that lysosomes are physiological stores of Fe(II) and that lysosomal impairment or dysregulation of their ability to acidify leads to perturbations in iron homeostasis, and so we hypothesize that LRRK2 mutations may dysregulate the dynamics of TfR transport to lysosomes and the integrity of the target lysosomal membranes, thus affecting lysosomal iron stores. This hypothesis warrants further investigation in the context of iron dysregulation in disease.

Iron deposition in the brain is a feature of PD and other neurodegenerative diseases. Imaging and biochemical methods have confirmed iron accumulation in the SN of PD patients

correlating with severity of motor symptoms [49–52]. Iron deposition can be detected by transcranial sonography in postmortem brains from PD patients, which correlates with increased ferritin levels and loss of neuromelanin content as assessed biochemically [53]. Studies on animal models have also supported a role of iron in nigral neurodegeneration in PD. In a model of acute MPTP intoxication in mice, dopaminergic cell loss correlated with iron accumulation and increase in lipoperoxidation coinciding with up-regulation of the divalent metal transporter DMT1 [54]. Furthermore, iron chelators can rescue dopaminergic neuron loss and behavioral effects caused by intracerebroventricular administration of 6-hydroxydopamine in rats [55]. In the case of LRRK2, higher nigral iron deposition has been reported in LRRK2 mutation carriers compared to idiopathic patients, while also the same trend has been highlighted in Parkin mutation carriers [56]. Pink1 and Parkin have been linked to degradation of mitochondrial iron importers [57,58] while iron overload can induce a Pink1/Parkin-mediated mitophagic response [59]. These data identify convergent pathways that regulate iron homeostasis and that may be involved in PD pathogenesis in patients.

Our data reflect altered iron handling in microglia in G2019S knock-in versus WT LRRK2 mice, but how this relates to the reported increase in iron in dopaminergic neurons, as relevant for PD pathogenesis, remains unclear. Studies have reported a cooperative effect of neuroinflammation and iron accumulation. Microglial activation by LPS induces secretion of IL-1β and TNF-α that in turn activate iron regulatory proteins in dopaminergic neurons inducing iron overload and neurotoxicity [60]. In these experiments, the iron status of microglia exacerbated proinflammatory cytokine release and neuronal degeneration. Furthermore, in mixed-culture models, microglia play a pivotal role in iron-elicited dopaminergic neurotoxicity via increase in cytokine production [61]. The iron storage protein ferritin is reportedly decreased in the SN of PD patients compared to controls [49]. Increased iron deposition together with lower ferritin levels could indicate increase in intracellular labile iron pools that constitute chelatable redox-active iron damaging to cells. Transgenic mice overexpressing human ferritin protein in dopaminergic SN neurons do not exhibit increases in reactive oxygen species or SN neuron loss following systemic administration of MPTP [62]. In G2019S LRRK2 knock-in mice, we observe an increase in ferritin that may represent a compensatory event to limit neurotoxicity, as a result of dysregulated Tf recycling and iron accumulation.

Iron overload in the brain can activate glial cells and promote the release of inflammatory and neurotrophic factors that control iron homeostasis in dopaminergic neurons. Increased iron in neurons can be toxic through production of hydroxyl radicals via Fenton chemistry, responsible for oxidation of lipids, proteins, and DNA. LRRK2 is modulated by oxidative stress in cells and PD-linked mutations compromise mitochondrial integrity [28,63,64]. Recent studies have demonstrated direct delivery of iron from Tf-endosomes to mitochondria via "kiss-and-run" events while highlighting how these events sustain mitochondrial biogenesis [65]. Dysregulation of iron storage or uptake by LRRK2 in microglia may in turn affect cytokine production and neuronal survival, as well as have a feedback effect on LRRK2 activity in the brain.

We have not yet established whether the observed effects of LRRK2 on iron homeostasis and transferrin recycling are damaging in the context of disease. This is due to the limitations of the available knock-in animal models that they do not present with baseline neurodegeneration. As a consequence of this, it remains uncertain whether iron modulation is a promising therapeutic avenue in the context of LRRK2 mutation carriers. Iron chelators have proven promising in neurodegeneration with brain iron accumulation disorders [66], but iron chelation therapy did not improve motor-UPDRS scores and quality of life significantly [67]. In recent years, LRRK2 has been nominated to play roles in a number of cellular processes ranging from inflammation, autophagy, and endolysosomal pathways to mitochondrial

homeostasis, processes that may also be cell type specific. Given the role that iron dyshomeostasis seems to play in basal ganglia diseases, a combination therapy of iron chelation along with inhibition of LRRK2 could present a viable strategy.

Our data suggest an effect of LRRK2 mutations on Rab8a function, driven by increased kinase activity, which may drive a dysfunction of iron uptake mechanisms in response to inflammatory stimuli in resident microglia. Deciphering the protein trafficking pathways around LRRK2 will help us understand the mechanistic underpinnings of neurodegeneration and the biological implications of blocking LRRK2 kinase in the clinic, while highlighting signaling avenues that can be targeted as therapeutic means.

# Materials and methods

## Cell culture, treatments, and constructs

HEK293FT cells (Thermo Scientific) were cultured in DMEM supplemented with 10% FBS and maintained at 37˚C, 5% CO2. HEK293T cell lines stably expressing different GFP LRRK2 variants were grown and cultured as described previously [68]. HEK293FT cells were transfected using Lipofectamine 2000 using standard procedures. For siRNAs transfection, cells were transfected with the SMARTpool ON-TARGETplus or scrambled siRNA control (Dharmacon) using the DharmaFECT reagent, according to the manufacturer's instructions. The 3×FLAG-tagged construct of LRRK2 in pCHMWS plasmid was a gift from Dr. J. M. Taymans (KU Leuven, Belgium). LLOMe treatment was at 1 mM for 4 hours, prior to fixation and staining. Nocodazole treatment was at 200 nM for 2 hours.

## Live imaging

Mouse primary astrocytes were transiently transfected with HaloTag-LRRK2(G2019S), GFP-Rab8a, and LAMP1-RFP using lipofectamine reagents. Cells were incubated with the JFX650 ligand (100 nM) for 1 hour, washed, and imaged using a Nikon SoRa spinning disk microscope utilizing 3D Landweber deconvolution, 48 hours later.

## Structure modeling

Heterodimeric complexes (Rab8a and Rabin8) and (Rab8a and Mical-L1) were modeled in PyMol (PDB: 4LHY and 5SZH, respectively) [69–71]. Insertion of the phosphate at T72 was modeled, and resulting distances were measured using the tools available (PyMOL version 2.0).

## Animal procedures

C57BL/6J mice were housed in standardized conditions at 2 to 5 animals per cage and with ad libitum access to food and water on a 12-hour light–dark cycle. The protocols used here are approved by the Institutional Animal Care and Use Committee of National Institute on Aging, NIH (protocol ID: 463-LNG-2021). The NIA IRP maintains an assurance with the Office of Laboratory Animal Welfare (OLAW) via the Office of Animal Care and Use (OACU) of the NIH. The NIA IRP complies with standards of the Guide for the Care and Use of Laboratory Animals, NRC, 2011, and the PHS Policy on Humane Care and Use of Laboratory Animals, USDHHS, NIH, OLAW, 2015 for all animals, as well as the Animal Welfare Act, USDA, regulations and USDA Animal Care Policies for USDA's Animal Welfare Act, USDA, APHIS.

## Stereotaxic surgery

One-year-old WT, LRRK2 KO, or LRRK2 G2019S mice were kept under anesthesia using 1% to 2% isoflurane. Mice were placed into a stereotaxic frame, an incision was made above the midline, and the skull was exposed using cotton tips. At anteroposterior +0.2 mm, mediolateral +/−2.0 mm from bregma (bilateral injection), a hole was drilled into the skull. A pulled glass capillary (blunt) attached to a 5-μl Hamilton glass syringe was used for injecting either 1 μl of either PBS or 5 mg/ml LPS solution (5 μg) per hemisphere. The capillary was lowered to dorsoventral −3.2 mm from bregma into the dorsal striatum. The solution was delivered at a rate of 0.1 μl per 10 seconds. After the injection, the capillary was held in place for 2 minutes, retracted 0.1 μm, and another 1 minute was waited before it was slowly withdrawn from the brain. The head wound was closed using surgical staples. Ketoprofen solution at 5 mg/kg was administered subcutaneously as analgesic treatment for the following 3 days.

## Histology

Animals were killed 3 days after surgery. Mice were deeply anesthetized using an IP injection of 200 μl of 10% ketamine. The thoracic cavity was opened to expose the heart. The whole body was perfused with 10 ml of 0.9% NaCl (2 minutes). Brains were removed, the left hemisphere was used for WB analysis, while the right hemisphere was fixed in 4% PFA for 48 hours. After 2 days, fixed hemispheres were transferred to 30% sucrose solution for cryoprotection. The brains were cut into 30 μm thick coronal sections—6 series—and stored in antifreeze solution (0.5 M phosphate buffer, 30% glycerol, 30% ethylene glycol) at −20°C until further processed.

## Immunohistochemistry

Sections were washed with PBS and incubated for 30 minutes in blocking buffer (10% Normal Donkey Serum (NDS), 1% BSA, 0.3% Triton in PBS). Afterwards, primary antibodies rabbit anti-FTH1 (D1D4; 4393S, Cell Signaling) and goat anti-Iba1 (Abcam, ab5076) were used at 1:500 and incubated overnight at 4°C in 1% NDS, 1% BSA, 0.3% Triton in PBS. Next day, sections were washed 3× for 10 minutes each with PBS and incubated with Alexa Fluorophore (568 or 647)-conjugated secondary antibodies for 1 hour at room temperature (RT). After 3 washes with PBS, sections were mounted on glass slides, coverslipped using Prolong Gold Antifade mounting media (Invitrogen), and imaged using a Zeiss LSM 880 confocal microscope equipped with Plan-Apochromat 63X/1.4 numerical aperture oil-objective (Carl Zeiss AG). Sections were further imaged using an Olympus VS120 (Olympus, Center Valley, Pennsylvania) slide scanner microscope.

## Perls blue staining

Sections were washed in ddH2O 3× for 10 minutes each. Afterwards, sections were incubated in a 1:1 mix of 4% potassium ferrocyanide (Sigma-Aldrich P3289-100G) and 4% HCl. After 30 minutes, brain slices were washed 3× for 10 minutes with PBS and quenched using 10% Methanol + 3% H2O2 diluted in PBS for 1 hour. Sections were then rinsed again using PBS and incubated in 3,3′-Diaminobenzidine and H2O2 according to instructions (SIGMAFAST D4418-50SET, Sigma-Aldrich). Then, slices were washed with PBS 3× for 10 minutes, mounted on SuperFrost Plus slides (Fisher Scientific), dried overnight and dehydrated using 70% Ethanol, 95% Ethanol, 100% Ethanol followed by Xylene and DPX mounting media (Sigma, 06522), and analyzed on ImageJ for signal intensity of selected ROIs.

## ICP-MS

Total iron concentrations in the samples were measured by ICP-MS (Agilent model 7900). For each sample, 200 μL of concentrated trace-metal-grade nitric acid (Fisher) was added to 25 μL or 100 μL of sample taken in a 15-mL Falcon tube. Tubes were sealed with electrical tape to prevent evaporation, taken inside a 1-L glass beaker, and then placed at 90˚C oven. After overnight digestion, each sample was diluted to a total volume of 4 mL with deionized water and then analyzed by ICP-MS.

## iPSC differentiation

iPSC lines were derived from reprogrammed peripheral mononuclear blood cells (PBMCs) collected from participants of the Parkinson Progression Marker Initiative (PPMI). Differentiation of iPSC to microglia was accomplished via a hematopoietic stem cell intermediate stage according to published protocols [72,73]. Mature iPSC-derived microglia were characterized by western blot and ICC for Iba1 expression (S6 Fig). The lines used were PPMI 3448 (WT) and PPMI 51782 (G2019S LRRK2 carrier). LPS activation was done in differentiation media at 100 ng/ml for 16 hours prior to fixation and staining. Activation by LPS treatment was verified by WB of pNFKb, p38, and imaging of Iba1 staining (S5 Fig).

## Co-immunoprecipitation

HEK-293T were transfected with 3×FLAG-Rab8a variants and 3xFLAG-GUS (Fig 3D) or 3xFLAG LRRK2 variants and GFP-Rab8a (Fig 3E), using Lipofectamine 2000 as per the manufacturer's instructions. After 24 hours, cells were lysed in buffer containing 20 mM Tris/HCl (pH 7.4), 137 mM NaCl, 3 mM KCl, 10% (v/v) glycerol, 1 mM EDTA, and 0.3% Triton X-100 supplemented with protease inhibitors and phosphatase inhibitors (Roche Applied Science). Lysates were centrifuged at $21,000 \times g$, 4˚C for 10 minutes, and the supernatants were analyzed for protein concentration (Pierce). About 5 μg of total protein from each supernatant was analyzed by SDS-PAGE for expression of the proteins in question. For FLAG immunoprecipitations, 3 mg of each sample was precleared with EZview protein G beads (Sigma-Aldrich) for 0.5 hours at 4˚C, and, subsequently, FLAG M2 beads were incubated with the lysates for 2 hours at 4˚C to IP target constructs. FLAG-tagged proteins were eluted in 1× kinase buffer (Cell Signaling), containing 150 mM NaCl, 0.02% Triton, and 150 ng/μl of 3xFLAG peptide (Sigma-Aldrich) by shaking for 30 minutes at 4˚C. For GFP IPs, Chromotek-GFP-Trap-agarose resin (Allele Biotech) was incubated with lysates for 2 hours at 4˚C. The beads were washed 4 times with buffer containing 20 mM Tris/HCl (pH 7.4), 137 mM NaCl, 3 mM KCl, and 0.1% Triton X-100. The washed beads were boiled for 6 minutes in 4× NuPAGE loading buffer (Invitrogen) supplemented with 1.4 M β-mercaptoethanol and analyzed by western blot.

## Cell surface biotinylation

Cells stably expressing GFP WT or G2019S LRRK2 were surface biotinylated, lysed, and surface proteins were enriched by avidin-conjugated agarose following the manufacturer's protocol (Pierce kit, A44390). Surface enriched proteins were analyzed by western blot.

## Western blot

Standard western blot protocols were used with the following antibodies: anti-LRRK2 (ab133474; Abcam), anti-Rab8a (6975, Cell Signaling), anti-MICAL-L1 (H00085377, Abnova),

GDI1/2 (716300, Thermo Scientific), Rabin8 (12321-1-AP, Protein Tech), FLAG (F1804, Sigma-Aldrich).

## RNA isolation and RNA-Seq

For Fig 5A–5D, RNA was isolated from primary microglia and analyzed by bulk RNA sequencing following methods published in [34]. For Fig 5E–5I, mice were administered striatal LPS injections and killed 3 days later. Collected brain hemispheres were dissociated using the Adult Brain Dissociation kit (Miltenyi Biotec), and brain microglia cells were isolated using CD11b microbeads and MACS sorting columns (Miltenyi Biotec). Single-cell suspensions were subjected to scRNA-seq at the National Cancer Institute Single-Cell Analysis Core Facility, Bethesda, Maryland, and followed the 10xGenomics pipeline. Data were analyzed using the R package Seurat (version 3, PMIDS: 26000488; PMID: 29608179) as described in [34].

## Immunocytochemistry

HEK293fT cells were seeded at $0.1 \times 10^6$ cells/well on 12 mm coverslips precoated with poly-D-lysine (Millicell EZ slide, Millipore) and cultured as described above. Cells were fixed in 4% (w/v) formaldehyde/PBS for 15 minutes, permeabilized in 0.2% Triton X-100/PBS for 10 minutes at RT, blocked in 5% (v/v) FBS in PBS, and incubated with primary antibodies in 1% (v/v) FBS/PBS for 3 hours. Following 3 washes in PBS, the cells were incubated for 1 hour with secondary antibodies (Alexa Fluor 488, 568, 647-conjugated; ThermoFisher). After 3 PBS washes, the coverslips were mounted, and the cells were analyzed by confocal microscopy (Zeiss LSM 880 and Airyscan superresolution microscopy). For high-content imaging, where Rab8a translocation was quantified, cells were plated in 96-well plates, precoated with matrigel, transfected with LRRK2 mutants, and processed following 24 hours of expression. Cells were treated with DMSO or 1 μM MLi-2 for 1 hour prior to fixation and staining with the corresponding antibodies. Cells were visualized with the ThermoFisher Cellomics ArrayScan using the HCS Studio platform and the SpotDetector v4 bioassay protocol. A minimum of 800 cells were imaged per well from a total of 6 wells per construct and condition, and Rab8a localization was analyzed. The ezColocalization ImageJ plugin was used for analysis of the Manders overlap coefficient [74]. Analysis of Rab8a presence near the centrosome was performed as in [30], where the percentage of Rab8a staining that resides within a 2.2-μm diameter circular ROI enclosing centrosomal staining, versus whole-cell Rab8a staining, was calculated. The antibodies used were Rab8a (Cell Signaling Technology; #6975), TfR1 (Abcam; ab84036), Ferritin Heavy Chain (Thermo Fisher; PA5-27500), transferrin (Abcam; ab82411), MICAL-L1 (Novus; H00085377-B01P), Cathepsin D (Calbiochem; 219361), Lamp1 (DSHB; 1D4B), and Lamp2 (DSHB; H4B4).

## Imaris analysis

Following superresolution microscopy, the Imaris platform (Bitplane, Zürich, Switzerland) was used to analyze the localization of TfR and Tf in different experiments. In Fig 4A, the Imaris Spots Detection module was used to identify TfR vesicles throughout z planes in the 3D volume, and the distance between them was calculated. In Fig 6, superresolution (Airyscan) z-stack images of iPSC-derived microglia were processed through the Imaris Surface Contour module to render Tf and nuclear staining to surfaces and measure the distance of Tf vesicles from the nucleus edge (3D rendering process outlined in S1–S4 Movies).

## Transferrin uptake and recycling assay

Cells that were plated and transfected on 12 mm poly-D-lysine coated coverslips were processed following 24 hours of expression. Initially, cells were incubated in prewarmed DMEM for 45 minutes. Alexa 568 fluorophore–conjugated transferrin (T23365, ThermoFisher) was added to the media at 20 μg/ml final concentration, and cells were incubated for 20 minutes. Following one quick wash in prewarmed media, cells were incubated with prewarmed DMEM, 10% (v/v) FBS media for 10 minutes and fixed and stained as detailed above. For the uptake assay, cells were plated on matrigel-coated 96-well plates (15,000 cells/well), transiently transfected and processed 24 hours posttransfection. Cells were incubated in DMEM for 45 minutes, and subsequently equal volume of DMEM media containing Alexa 568–conjugated transferrin was added to the wells to achieve 20 μg/ml final transferrin concentration. Transferrin was added at different time intervals, and the plate was washed in cold PBS and fixed at 20 minutes of total uptake, achieving 0, 2, 4, 6, 8, 10, 15, and 20 minutes of uptake time points. A similar setup was followed for recycling assays whereby cells were loaded with Tf-568 in DMEM for 30 minutes, washed once in DMEM, and then incubated in DMEM/10% FBS media for the different time points of recycling, followed by a quick wash in cold PBS and fixation. Similar paradigms were followed in Tf uptake/recycling assays in iPSC-derived microglia, using DMEM/F12 media plus supplements as per differentiation protocols.

## Statistical analysis

Statistical tests used are noted in figure legends of representative graphs. Briefly, one-way ANOVA or two-way ANOVA with Tukey post hoc test and two-tailed unpaired Student $t$ test with Mann–Whitney $U$ post hoc test were used. All statistical analyses were performed using GraphPad Prism 7 (GraphPad Software, San Diego, California).

## Supporting information

**S1 Fig. Validation of Rab8a antibodies for endogenous detection.** (A, B) The rabbit anti-Rab8A antibody by Cell Signaling (D22D8; #6975) and rabbit anti-Rab8a (#ab188574) antibody were tested in ICC and western blotting against Rab8a siRNA treated Hek293T cells. The Cell Signaling (#6975) antibody validated against knock-down for detection of endogenous Rab8a and was used throughout this study. ICC, immunocytochemistry; siRNA, small interfering RNA.
(TIF)

**S2 Fig. LRRK2 genetic variants sequester endogenous Rab8a to lysosomes in a kinase-dependent manner.** (A) HEK293T cells transiently expressing WT or mutant LRRK2 constructs were treated with MLi-2 or DMSO for 1 hour prior to fixation and staining for Rab8a, Lamp2, and LRRK2 (FLAG). Rab8a was sequestered to Lamp2-positive lysosomes in cells expressing R1441C, Y1699C, G2019S, or I2020T LRRK2, but not the kinase-dead K1906M LRRK2 or transfection control (GUS). MLi-2 treatment rescues this phenotype. (B) High-content imaging (Cellomics) was used to quantitate the percentage of cells exhibiting sequestered Rab8a ($>4$ μm$^2$ Rab8a foci, $>800$ cells imaged per well, 4 wells per condition). Higher Rab8a sequestration was observed in cells expressing R1441C, Y1699C, G2019S, and I2020T LRRK2 compared to WT LRRK2, while this is ameliorated by MLi-2. (C) The percentage of cells that exhibited Rab8a-positive lysosomes was significantly increased in cells expressing R1441C LRRK2 or G2019S LRRK2 compared to WT LRRK2 expressing cells (a minimum of 200 cells were counted per group across $N = 4$ different experiments, one-way ANOVA Tukey post hoc, $^{***}P < 0.001$, $^*P < 0.05$, F (2, 9) = 17.03). The underlying data can be found in S1 Data.

LRRK2, leucine-rich repeat kinase 2; WT, wild-type.
(TIF)

**S3 Fig. Association of Rab8a with centrosomes and lysosomes in HEK293T cells expressing G2019S LRRK2.** (A) HEK293T cells transiently expressing FLAG G2019S LRRK2 were treated with nocodazole (200 nM) or DMSO for 2 hours and stained for FLAG G2019S LRRK2, endogenous Rab8a, and pericentrin. (B) Quantification of the percentage of Rab8a staining that resides within a 2.2-μm diameter circular ROI enclosing centrosomal staining versus whole-cell Rab8a staining (B: $^{**}P < 0.01$, $N > 19$ cells per group from 2 independent experiments, two-tailed Student $t$ test, Mann–Whitney $U$). (C) Cells were treated as in (A) and stained for FLAG G2019S LRRK2, endogenous Rab8a, and endogenous Lamp2. (D) Quantification of Manders colocalization coefficient between Rab8a and lamp2 staining (D: $^{**}P < 0.01$, $N > 14$ cells per group from 2 independent experiments, two-tailed Student $t$ test, Mann–Whitney $U$). The underlying data can be found in S1 Data. LRRK2, leucine-rich repeat kinase 2; ROI, region of interest.
(TIF)

**S4 Fig. Endogenous MICAL-L1 is recruited by mutant LRRK2 to lysosomes.** (A) HEK293T cells transiently expressing WT or mutant LRRK2 were stained for MICAL-L1, LRRK2 (FLAG), and Lamp2 and analyzed by confocal microscopy. The Manders overlap coefficient between MICAL-L1 and LRRK2, or MICAL-L1 and Lamp2 were calculated in ImageJ. (B,C: $N > 10$ cells per group collected in 1 experiment; one-way ANOVA Tukey post hoc, $^{***}P < 0.001$, $^{**}P < 0.01$, $^{*}P < 0.05$, B: F $(2, 30) = 13.66$, C: F $(2, 31) = 11.40$). The underlying data can be found in S1 Data. LRRK2, leucine-rich repeat kinase 2; WT, wild-type.
(TIF)

**S5 Fig. Mutant LRRK2 induces mistrafficking of TfR.** (A) HEK293T cells transiently expressing WT or mutant LRRK2 constructs were stained for endogenous TfR and Lamp2 and visualized by confocal microscopy. Colocalization between TfR and Lamp2 was analyzed in ImageJ. (A, B) ($N > 19$ cells per group collected in 2 independent experiment, $^{****}P < 0.0001$, one-way ANOVA with Tukey post hoc; F $(2, 42) = 30.92$). (C) Cells stably expressing WT LRRK2 were treated with LLOMe for 2 hours and stained. LLOMe treatment induced the sequestration of Rab8a to LRRK2-positive vesicles with partial recruitment of TfR. (D) HEK293T cells transiently expressing WT or G2019S LRRK2 were treated with MLI-2, loaded with Alexa Fluor 568-Tf and visualized at T10 minutes of recycling in fresh media containing MLi-2. The mean distance between Tf vesicles was calculated in Imaris ($N > 2,500$ vesicles were counted in at least 20 cells per construct from 2 independent experiments, $^{****}P < 0.0001$, one-way ANOVA with Tukey post hoc, F $(2, 7932) = 560.1$). (E) Cells stably expressing GFP WT or G2019S LRRK2 were surface biotinylated and surface proteins were enriched by avidin-conjugated agarose before western blotting for TfR ($N = 2$ experimental replicates). The underlying data can be found in S1 Data. LRRK2, leucine-rich repeat kinase 2; TfR, transferrin receptor; WT, wild-type.
(TIF)

**S6 Fig. Characterization of iPSC-derived microglia.** (A) Immunoblot analysis of iPSC-derived microglia following treatment with 100 ng/mL LPS. (B, C) Quantification of indicated phospho-proteins normalized against total protein and loading control b-actin in treated vs. untreated cells (technical $N = 3$, unpaired $t$ test, $P < 0.01$). (D) Iba1 staining of DMS and LPS treated microglia from WT and G2019S cell lines. (E) Cell size of iPSC-derived microglia under DMSO or LPS treatment was measured in ImageJ ($N > 19$ cells counted in 2 differentiations). The underlying data can be found in S1 Data. iPSC, induced pluripotent stem cell; LPS,

lipopolysaccharide; WT, wild-type.
(TIF)

**S7 Fig. Tf recycling in iPSC-derived microglia.** iPSC-derived human microglia from WT or
G2019S LRRK2 carriers were treated with LPS, and high-content imaging was used to monitor
the uptake of Alexa Fluor 568–conjugated Tf (A, B) or recycling following Tf preloading in
pulse-chase experiments (C, D). ($N$ = 5 technical replicates of >50 counted cells, two-way
ANOVA; B: time point: $P < 0.0001$, F (4, 80) = 155.1, genotype: NS; D: time point: $P < 0.0001$,
F (4, 80) = 115.2, genotype: $P < 0.05$, F (3, 80) = 3.664). The underlying data can be found in
S1 Data. iPSC, induced pluripotent stem cell; LPS, lipopolysaccharide; LRRK2, leucine-rich
repeat kinase 2; Tf, transferrin; WT, wild-type.
(TIF)

**S8 Fig. Uncropped western blots of figures.** (A) Panel of Fig 3A: HEK293T cells expressing
control (GUS), FLAG WT, or mutant LRRK2 variants were treated with MLi-2 prior to lysis
and analysis by western blotting for pS1292 and total LRRK2, as well as pT72 and total Rab8a.
(B) Panel of Fig 3D: HEK293T cells expressing FLAG WT, I2020T, or K1906M LRRK2 along
with GFP Rab8a, GFP control, or FLAG-GUS control were lysed and proteins immunoprecipi-
tated using GFP beads. Co-immunoprecipitated proteins were analyzed by western blotting
probed for MICAL-L1 as well as pT72 and total Rab8a. (C) Panel of S5E Fig: Cells stably
expressing GFP WT or G2019S LRRK2 were surface biotinylated, and surface proteins were
enriched by avidin-conjugated agarose before western blotting for TfR. (D) Panel of S6A Fig:
Immunoblot analysis of iPSC-derived microglia following treatment with 100 ng/mL LPS. (E)
Panel of S1 Fig: western blot validation of Rab8a antibodies for endogenous detection. iPSC,
induced pluripotent stem cell; LPS, lipopolysaccharide; LRRK2, leucine-rich repeat kinase 2;
TfR, transferrin receptor; WT, wild-type.
(TIF)

**S1 Data. Individual numerical values that underlie the summary data displayed in the fol-
lowing figure panels: Figs 1B, 1C, 1F, 2B, 3B, 4B, 4D–4G, 4J, 4K, 6B–6F, 7C, 8B, 8C, S2B,
S2C, S3B, S3D, S4B, S4C, S5B, S5D, S5E, S6B, S6C, S6E, S7B and S7D.**
(XLSX)

**S1 Movie. Imaris 3D surface rendering of endogenous transferrin and Lamp2 staining in
WT LRRK2 iPSC-derived microglia.** iPSC, induced pluripotent stem cell; LRRK2, leucine-
rich repeat kinase 2; WT, wild-type.
(MP4)

**S2 Movie. Imaris 3D surface rendering of endogenous transferrin and Lamp2 staining in
WT LRRK2 iPSC-derived microglia treated with 100 ng/ml LPS for 16 hours.** iPSC,
induced pluripotent stem cell; LPS, lipopolysaccharide; LRRK2, leucine-rich repeat kinase 2;
WT, wild-type.
(MP4)

**S3 Movie. Imaris 3D surface rendering of endogenous transferrin and Lamp2 staining in
G2019S LRRK2 iPSC-derived microglia.** iPSC, induced pluripotent stem cell; LRRK2, leu-
cine-rich repeat kinase 2.
(MP4)

**S4 Movie. Imaris 3D surface rendering of endogenous transferrin and Lamp2 staining in
G2019S LRRK2 iPSC-derived microglia treated with 100 ng/ml LPS for 16 hours.** iPSC,

induced pluripotent stem cell; LPS, lipopolysaccharide; LRRK2, leucine-rich repeat kinase 2. (MP4)

## Acknowledgments

We thank the NCI-CCR Single Cell Analysis Facility, Cancer Research Technology Program at the Frederick National Lab for Cancer Research for critical assistance with single cell RNA-Seq analysis. We thank Dr. George R. Heaton (Icahn School of Medicine at Mount Sinai) for critical discussions on experimental procedures and data analysis.

## Author Contributions

**Conceptualization:** Adamantios Mamais, Ravindran Kumaran, Changyoun Kim, Mark R. Cookson.

**Data curation:** Luis Bonet-Ponce, Nathan Smith, Alexandra Beilina, Alice Kaganovich, Manik C. Ghosh, Rina Bandopadhyay, David C. Gershlick, Mark R. Cookson.

**Formal analysis:** Adamantios Mamais, Jillian H. Kluss, Luis Bonet-Ponce, Natalie Landeck, Rebekah G. Langston, Nathan Smith, Alexandra Beilina, Alice Kaganovich, Manik C. Ghosh, Laura Pellegrini, Ravindran Kumaran, Ioannis Papazoglou, George R. Heaton, Nunziata Maio.

**Funding acquisition:** Mark R. Cookson.

**Investigation:** Adamantios Mamais, Jillian H. Kluss, Natalie Landeck, Rebekah G. Langston, Alexandra Beilina, Laura Pellegrini, Nunziata Maio, Changyoun Kim, David C. Gershlick.

**Methodology:** Adamantios Mamais, Changyoun Kim.

**Project administration:** Mark R. Cookson.

**Resources:** Matthew J. LaVoie.

**Supervision:** Matthew J. LaVoie, Mark R. Cookson.

**Writing – original draft:** Adamantios Mamais, Mark R. Cookson.

**Writing – review & editing:** Mark R. Cookson.

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
