## [Editor Report · Decision Letter 0]

16 Sep 2020

Dear Dr Mamais, 

Thank you for submitting your manuscript entitled "Pathogenic mutations in LRRK2 sequester Rab8a to damaged lysosomes and regulate transferrin-mediated iron uptake in microglia" for consideration as a Research Article by PLOS Biology. Please accept my apologies for the delay in sending this initial decision to you.

Your manuscript has now been evaluated by the PLOS Biology editorial staff, as well as by an academic editor with relevant expertise, and I am writing to let you know that we would like to send your submission out for external peer review.

Please re-submit your manuscript within two working days, i.e. by Sep 18 2020 11:59PM.

Kind regards,

Gabriel Gasque, Ph.D.,

Senior Editor

PLOS Biology

---

## [Decision Letter · Decision Letter 1]

23 Oct 2020

Dear Dr Cookson,

Thank you very much for submitting your manuscript "Pathogenic mutations in LRRK2 sequester Rab8a to damaged lysosomes and regulate transferrin-mediated iron uptake in microglia" for consideration as a Research Article at PLOS Biology. Your manuscript has been evaluated by the PLOS Biology editors, by an Academic Editor with relevant expertise, and by three independent reviewers. Please accept my apologies for the delay in sending the decision below to you.

The reviews of your manuscript are appended below. You will see that the reviewers find the work potentially interesting. However, based on their specific comments and following discussion with the Academic Editor, I regret that we cannot accept the current version of the manuscript for publication. We remain interested in your study and would be willing to consider resubmission of a comprehensively revised version that thoroughly addresses all the reviewers' comments. We cannot make any decision about publication until we have seen the revised manuscript and your response to the reviewers' comments. Your revised manuscript would be sent for further evaluation by the reviewers.

The reviewers agree that the idea that LRRK mutations cause iron misregulation, and that this occurs via sequestration of Rab8a in lysosomes and dysregulation of TfR trafficking, is of potential interest. However, both Reviewer 1 and 2 feel that a substantial amount of additional experimental support would be required to solidify these conclusions.

We appreciate that addressing these requests with new experiments represent a great deal of extra work, and we are willing to relax our standard revision time to allow you six months to revise your manuscript.

**IMPORTANT - SUBMITTING YOUR REVISION**

*Resubmission Checklist*

*Published Peer Review*

*PLOS Data Policy*

*Blot and Gel Data Policy*

Sincerely,

Gabriel Gasque, Ph.D.,

Senior Editor,

ggasque@plos.org,

PLOS Biology

REVIEWS:

Reviewer #1: In this manuscript, the authors conclude that mutant LRRK2 sequesters Rab8a in lysosomes and dysregulates TfR trafficking, leading to iron accumulation in brain striatal microglia of G2019S LRRK2 mice. This is an important area of broad interest to the readers of PLoS, but as discussed in detail below, the present story would require significant additional work before it could be considered for presentation in PLoS Biology; the conclusions are premature.

First the authors conclude that pathogenic LRRK2 causes Rab8a to be localized in lysosomes. Hilfiker has shown that Rab8a vesicles are concentrated near the centriole and lysosomes may also accumulate at the same location as independent structures. Proof of co-localization would require addition of a microtubule depolymerizing drug such as nocodazole to be sure that separate structures were not simply near each other. What fraction of total Rab8a co-localizes with a lysosomal marker ± nocodazole? The authors use a metric, "cells with sequestered Rab8a vesicles", monitoring the number of Rab8a vesicles per cell. At this level of resolution they are not really measuring vesicle numbers as the vesicles are below the resolution limit of this type of microscopy. Also of concern is the fact that when two adjacent cells are shown, they do not show the same phenotype as the cells being outlined. Nevertheless, a more appropriate metric would be percent of cell area occupied by labeled pixels in the various conditions to indicate possible concentration. In E, the authors detect a Rab8a decorated lysosome—and in D use arbitrary units to document co-localization between Rab8a and LRRK2. What percent of pixels stained for each marker co-localize with the other marker and vice versa? "High throughput colocalization analysis" doesn't match the data shown so the authors need to be more hands on in their analysis and cells need to be viewed and analyzed at high magnification (100X lens), unclustered. A single Airyscan image shows Rab8a lysosome decoration but in their prior work, they reported that in the absence of LLOME, this was only seen for a very few percent of total lysosomes (ref 17). 

Figure 2. Here, the authors report what they have already published—Rab8a on LLOME lysosomes. Line scans are used to document lack of co-localization of cathepsin D with LAMP2 in LRRK2 expressing cells. A line scan is a poor mode of reporting co-localization as all data not under the line is lost. Here, again, it is essential to ask what fraction of total label co-localizes.

Next the authors confirm that pRab8a cannot bind Rabin8 but binds MICAL-L1. This confirms a report by Steger., 2017 using mass spec IP, and is consistent with the colocalization in cells shown here ±LRRK2.

Figure 4 shows an apparent coalescence or local concentration of TfR containing membranes. These are not necessarily lysosomes, they are simply concentrated in one cell region and would require a nocodazole experiment to determine if they are actually the same membrane compartment as lysosomes. Again, because the vesicles are not resolved, area occupied by TfR might be more revealing? In 4E, the authors imply that cells take up more transferrin over time. This could be because they are slower at recycling the transferrin back into the medium or they take it up for a longer period of time. Experiments need to be included that monitor recycling (chase out the label and monitor decrease in fluorescence) and also quantify TfR surface numbers at steady state to provide a meaningful answer related to TfR function upon LRRK2 expression. In addition, the effect of a LRRK2 inhibitor on a single mutant cell type would need to be included for the results to be meaningful and know that differences were due to LRRK2.

Next the authors detect an increase in iron accumulation. If TfR mediated, this could only happen if there was more recycling to increase the "bucket brigade" of TfRs bringing in iron; this would be opposite of what one would predict from Rab phosphorylation? Again this is why the authors need to better characterize the entire transferrin cycle.

In Fig. 5 the authors used their previous RNA seq data and found lots of endolysosome hits correlating with inflammation but this is not a surprise and has been reported broadly elsewhere. They injected wild type or LRRK2 KO mice with LPS and looked at microglia; KO animals had fewer microglia but that could be due to independent pathways and does not reveal much. Next they looked at transferrin containing compartments in microglia ± mutant LRRK2 ± LPS; the structures and overlap is not meaningful without nocodazole to resolve the overlapping labeling, and overlap at steady state may not be indicative of trafficking differences.

 Finally, the authors saw significant differences in iron accumulation in G2019S brains—this may be far downstream of TfR due to an indirect lysosomal defect.

Reviewer #2: The manuscript by Mamais and colleagues describes effects of pathogenic LRRK2 on transferrin receptor recycling and iron accumulation in microglia. Whilst generally an interesting manuscript which warrants publication in PLoS Biology, I have several comments which need addressing as outlined below:

Major:

1. The authors validate the Rab8a-specific antibody (Figure S1) which they subsequently use to show accumulation of Rab8 on dysfunctional lysosomes in the presence of pathogenic LRRK2 expression. For validation purposes, they show an absence of Rab8a signal upon RNAi of Rab8a by ICC with the D22D8 antibody, and a knockdown of Rab8a protein as assessed by Western blotting. Please also show successful knockdown of Rab8a by Western blotting for the other two antibodies (ab128022 and ab188574) which do not show a decrease in staining upon RNAi of Rab8a (Figure S1). As indicated in materials and methods, ICC was performed in the absence of detergent. Given that the ab188574 antibody has been reported to be specific for Rab8a as assessed by Western blotting using Rab8a KO cell lysates, the authors should also evaluate whether the three antibodies shown in Figure S1 display specific staining in the presence of detergent.

2. The authors show that pathogenic LRRK2 expression causes lysosomal damage, associated with the accumulation of Rab8a and LRRK2 on dysfunctional, enlarged lysosomes (Figure 1). These findings are in contrast to other studies suggesting that chemical induction of lysosomal damage is required to trigger accumulation of LRRK2 and Rabs on enlarged, dysfunctional lysosomes (Eguchi et al., PNAS, 2018). The authors should discuss their findings in the context of other published data. In addition, they show reversal of Rab8a/LAMP2 colocalization and Rab8a/LRRK2 colocalization by application of LRRK2 kinase inhibitor MLi2 for 1 h. Is the observed lysosomal enlargement also reverted?

3. The authors show (Figure 3) colocalization of Rab8 with MICALL1 in cells expressing I2020T mutant LRRK2, but not wt LRRK2. The data presented here are not convincing, since Rab8a and MICALL1 also seem to colocalize in wildtype LRRK2 expressing cells. The authors should perform similar experiments as shown in Figure 1, determining colocalization of Rab8a with MICALL1, MICALL1 with LRRK2, and MICALL1 with LAMP2. The latter will confirm whether MICALL1 indeed is recruited to damaged lysosomes in the presence of pathogenic, but not wildtype LRRK2 expression.

4. The authors describe clustering of TfR-positive vesicles in cells overexpressing pathogenic LRRK2 (Figure 4). Please show whether this is reverted by application of LRRK2 kinase inhibitor MLi2. Importantly, they show that these structural changes are associated with increased sequestration of fluorescently-labelled internalized transferrin. Is this reverted by MLi2?

5. Whilst expression of all pathogenic LRRK2 mutants correlates with clustering of TfR-positive vesicles and an increase in internalized transferrin levels over time, the ICP-MS determination (Figure 4I) only reveals increased iron levels in cells expressing G2019S LRRK2. How do the authors explain this discrepancy?

6. Figure 5: the authors show RNA-seq data of wildtype primary microglia treated with either LPS of preformed a-syn fibrils in culture, and mention that the shared hits betweeen the two treatments reveal enrichment for endosomal and lysosomal pathways. Figure 5C shows the top hits related to endosomal (not lysosomal) pathways, so this statement in the result section is confusing. Similarly, they show single cell RNA-seq data from isolated microglia of wildtype versus LRRK2 KO mice upon either PBS or LPS injection. They show differences between resting and activated microglia in both cases, and related to various processes (Figure 5H). Are the genes depicted in Figure 5I the top hits (eg FTH1), or how do these genes relate to the gene ontology analysis of Figure 5H?

7. The authors use iPS-derived microglia from control and G2019S LRRK2 carriers to show clustering of TfR-positive vesicles which display increased size, but the relevance of these observations for TfR-mediated iron uptake and accumulation is not shown. Importantly, the authors should determine whether the structural observations correlate with altered internalization of fluorescently-labelled transferrin in these cells as well, similar to what they observed when overexpressing LRRK2 in HEK cells (Figure 4). Also, does LPS treatment of iPS-derived microglia in vitro alter protein levels of TfR or FTH1?

8. The authors show increased iron deposition in microglia in G2019S knockin mice as compared to controls upon intrastriatal LPS injection, associated with increased levels of FTH1 and Tf (Figure 7 and Figure 8). What about differences in iron levels in the absence of LPS injection, especially in the SN? Their data indicate that LPS treatment causes increased iron deposition in activated microglia in G2019S mice as compared to control. This may reflect altered iron handling in microglia in G2019S versus control mice (perhaps due to differences in microglial activation), but how this relates to the described increases in iron in dopaminergic neurons, as relevant for PD pathogenesis, remains unclear. In this context, the discussion would benefit from restructuring as well, especially with respect to the relevance of iron deposition in PD brains, as the authors do not provide evidence of altered iron handling in SN, in the absence of LPS, in the G2019S knockin mice as compared to control mice.

Minor:

1. please mention time and concentration of LLOME employed.

2. Results: the authors mention "Given the above data showing that LRRK2 phosphorylation prevents Rab8a activation and redirects Rab8a and MICAL-L1 away from the ERC,...". What evidence do the authors have that phosphorylation prevents Rab8a activation? If based on differential interactions of Rab8a mutants, this statement should be deleted, since these mutants have been shown not to mimic the phospho-status of the Rab protein.

Reviewer #3: Overall: The authors present an elegant set of studies demonstrating that a) LRRK2 GOF mutations lead to kinase-dependent sequestration of Rab8a in lysosomes, b) GOF mutations dysregulate iron transport by driving the iron-binding transporter transferring into Rab8b-containing lysosomes. The authors go on to show that microglia derived from iPSCs from G2019S heterozygotes display the transferrin dysregulation when exposed to inflammatory stress, consistent with the phenotype observed in LPS-treated G2019S KI mice. Technically the approaches are sound and the results are robust. The significance of these studies is high given that the field has maintained that LRRK2 function in brain microglia is limited and unimportant; yet these data underscore its role in stress responses, a role inherent in the fact that LRRK2 is a member of the RIP kinase family and these are regulators of cellular stress.

---

## [Decision Letter · Decision Letter 2]

14 May 2021

Dear Dr Cookson,

Thank you very much for submitting a revised version of your manuscript "Pathogenic mutations in LRRK2 sequester Rab8a to damaged lysosomes and regulate transferrin-mediated iron uptake in microglia" for consideration as a Research Article at PLOS Biology. This revised version of your manuscript has been evaluated by the PLOS Biology editors, by the Academic Editor, and by the original reviewers 1 and 2.

In light of the reviews (below), we will not be able to accept the current version of the manuscript. However, we are willing to grant a second and last chance for you to address the lingering concerns raised by the reviewers. We think this second revision needs to be experimental and the issues identified by the reviewers would need to be thoroughly and convincingly addressed for us to move forward with the work. It would be very unfortunate to have to reject the study after such a protracted process. Thus, we would understand if you decide to take your submission elsewhere at this point.

As you will see, we asked reviewer 2 to comment on reviewer 1's feedback. Reviewer 2 identified two concerns that would need further experimentation: "I feel that some of the comments from referee 1 could easily be addressed using another cell type transiently transfected with tagged LRRK2 constructs" and "A better way to address the nocodazole comment would be to employ a live imaging approach." We expect the revision to include these data.

We cannot make any decision about publication until we have seen the revised manuscript and your response to the reviewers' comments. Your revised manuscript is also likely to be sent for further evaluation by the reviewers.

We expect to receive your revised manuscript within 3 months. 

**IMPORTANT - SUBMITTING YOUR REVISION**

*Re-submission Checklist*

*Published Peer Review*

*PLOS Data Policy*

*Blot and Gel Data Policy*

Sincerely,

Gabriel Gasque

Senior Editor

PLOS Biology

ggasque@plos.org

REVIEWS:

Reviewer #1: Unfortunately, despite the language of the response letter, the corrections made by the authors do not adequately address the concerns raised in the original review. A paper in PLoS Biology should show the molecular basis for events described, not only possible morphological rearrangements. Unfortunately this manuscript does not do that and there are numerous errors throughout.

To follow are many comments that need to be addressed. Figure 1A. Where is Rab8 in the wild type and MLi-2 treated cells? Rab8 is broadly distributed as shown in papers from many other labs and these images are overcorrected such that most of the Rab8 signal is no longer detected. The images don't look right. Row 4, why is the cell with poor LAMP2 staining selected? Same for E and G--cells have much more Rab8a, and nocodazole should redistribute it, not make it disappear. That it does disappear is a strong indication that the images have been overcorrected. In 1E, why is the red cell with a lot of LRRK2 staining not examined? Why do they include the totally nonspecific 188574 Rab8 antibody in their analyses (Fig. S1)? Are the effects only seen in 293T cells upon massive overexpression of LRRK2?

Fig. S4. The purpose of nocodazole is to help demonstrate co-localization--that two markers move together because they are in the same compartment. The authors' own data show loss of colocalization upon nocodazole, which demonstrates that the markers are in DIFFERENT compartments, not that microtubule based motors are needed to move things.

"The interaction of Rab8a with the GDI was blocked by T22N and enhanced by Q67L." GDI does not bind at all to GTP-bearing Rabs so this is the opposite of what a huge body of work tells us should have been seen. Indeed, the figure 3 shows the opposite; this reviewer is surprised they saw any binding to Q67L. As for MICAL-L1 binding, it should have strongly enhanced by Q67L mutation and isn't so not sure what to conclude from data--perhaps it is because the IPs were done in EDTA with no magnesium or nucleotide. 

Transferrin uptake -- the authors have not addressed the issues raised. If they want to conclude there are differences in uptake, they need to explain how and why. More surface receptors? They say no. More recycling of receptors to the surface? They should follow the release of Tf in the same experiment as the uptake to be sure there are no differences in cell number or confluency that could differ between lines and change appearance of different mutants since the differences are so small.

Abstract - "Here, we show that gain-of-function mutations in LRRK2 induce sequestration of endogenous Rab8a into lysosomes in cells". The authors have not shown Rab8 sequestration as proven by lack of Rab8/LRRK2/LAMP colocalization in the presence of nocodazole, nor have they shown Rab8a to be "into" the lysosome. If Rab8a is in the lysosome, it has other implications.

There is inconsistency in cell outlines between figures, some have and others don't have them. figures 1A cell outlines don't match the actual cell staining.

Figure legends - The figure legends need to clearly explain what the figures actually describe not the results. Terms such as "cytocenter" in Fig1 legend are not explained. Legends of most figures (as well as the main text) need to indicate the type of cells used for each experiment. The number of cells and vesicles counted have been noted, but not whether standard errors from multiple experiments have been plotted. 

Vesicle quantitation - Pictures in Fig 6 do not reflect the colocalization numbers quantified with Pearson's (40% colocalization). Distribution of the vesicles and their relative distance from each other are also a function of the size and spreading of the cell. A smaller cell would naturally have more "clustered" vesicles as is seen in Fig 6A +/- LPS. Are these distances normalized for cell size?

Fig 3D/E - Have the authors looked at MICAL or MICAL-L1 in their western blots? These are two different proteins. The blots need to be quantified and replicates plotted to show (no) changes in interactions after immune precipitation. Some of the signals shown don't make any sense.

In Fig 1E/G, nocodazole treatment did not change perinuclear Rab8 accumulation. In S4 nocodazole treatment also dispersed TfR, LAMP and LRRK2. These data indicate that perinuclear clustered Rab8a, TfR, LAMP and LRRK2 are all on different vesicles, possibly in proximity until treated with nocodazole. This could explain why there is no difference in transferrin recycling or uptake in most of the experiments. Peri-nuclear transferrin / transferrin receptor accumulation is well known irrespective of pathogenic mutant LRRK2 activity (Maxfield et al 1984 Cell).

"Mutant LRRK2 expressing cells sequestered Tf in enlarged vesicles that were labelled with LRRK2 and Rab8a (Fig 4D) " - Sequestration cannot be concluded since data indicating transferrin being held in the perinuclear LRRK2-compartment longer for mutant than WT is not shown.

"ICP-MS analysis revealed significantly elevated iron levels in cells stably-expressing G2019S LRRK2 compared to WT LRRK2 (Fig 4K)". I2020T has the strongest effect on TfR according to authors, yet iron accumulation is unaltered. Moreover an MLi-2 or kinase inhibitor experiment was not performed. What would the molecular basis be for the data in 4K anyway?

Fig 7/8 showing increased iron accumulation (along with Tf and Fth) post LPS stimulation in striatal regions is potentially novel and interesting, more so because of the enhanced effect in G2019S and decreased effect in KO mice. The authors tried looking at striatal microglia and found no changes in Fth1 nor Iba1 (Fig 8E/D). Neither the data nor the images support this statement - 

"striatal microglia accumulate higher levels of ferritin-bound iron upon inflammation by LPS compared to WT LRRK2 mice". This increase could be in other cell types though. The levels of transferrin receptor and dmt1 are very low in various cell types in brain. 

Reviewer #2: 

Original comments: The authors have addressed all my comments, and I deem the manuscript suitable for publication in PLoS Biology in its present form.

After reading reviewer 1's feedback: I´ve carefully looked over the comments from referee 1 to both the initial and revised version of the manuscript. I think the present work is important, as revealing a link between pathogenic LRRK2 and deregulated iron handling, with the TfR likely accumulating in an endolysosomal compartment. In my opinion, the data in iPSC-derived microglia as well as in the G2019S knockin mouse model are of particular interest. 

 The bulk of the mechanistic work has been performed in HEK293 cells overexpressing pathogenic LRRK2, and often employing high-content imaging. I think this is where most of the critiques from referee 1 are coming from. HEK cells have a very small cytoplasm, such that any colocalization analysis becomes limited. The authors conveniently used HEK cells stably overexpressing wildtype and various LRRK2 mutants. I feel that some of the comments from referee 1 could easily be addressed using another cell type transiently transfected with tagged LRRK2 constructs and suited for quantitative high-resolution confocal image analysis. Some key experiments could be repeated to corroborate the author´s conclusions, and this would not entail a lot of additional work. 

 Apart from this, I think there are two main issues. One is related to nocodazole treatment (an experiment proposed by referee 1 in the first round of comments). I was not surprised to see the data presented, as phospho-Rab8 is not only localized to lysosomes, but also stably associates with centrosomes, and we have shown that this interaction is not disrupted by nocodazole. So there are likely at least two pools of phospho-Rab8, one bound to the centrosome, and the other to lysosome-related structures. Importantly though, most permeabilization conditions cause rapid extraction of prenylated Rab proteins from membranes, which makes interpretation of such experiments difficult. A better way to address the nocodazole comment would be to employ a live imaging approach. 

 The second main issue is related to the reported perinuclear TfR clustering, and how this may relate to TfR trafficking alterations. Some of my comments to the initial version of the manuscript were related to this as well. In the revised version of the manuscript, the authors try to address this issue by performing uptake/recycling assays, and the data presented indicate very subtle alterations. One could argue that the effects observed do not warrant a link between altered TfR trafficking and pathogenic LRRK2. Several years ago, we looked at TfR trafficking in pathogenic LRRK2 expressing cells and only found subtle alterations. I feel that the data presented in this manuscript properly reflect what is going on, namely a subtle effect on TfR trafficking which is hard to dissect (uptake versus recycling versus surface levels of receptors). The question is whether this is enough to give rise to the altered iron handling as observed in vivo. I feel that the authors shy away from making strong statements related to this mechanistic link, even though a potential link is obviously inferred. 

Other comments from referee 1 could be addressed by rephrasing the manuscript, for example: 

-mutations in Rab8a and differential binding to GDI or MICAL-L1: in some cases, these mutations do not properly mimic the GTP-bound versus GDP-bound version of a Rab protein. The data presented relating to MICAL-L1 have limited meaning, are not relevant for the conclusions of the manuscript, and could be removed altogether 

-abstract mentions "into" the lysosome, whilst they show recruitment of both proteins (LRRK2 and Rab8a) to the membrane of LAMP1-positive lysosomes. Careful rephrasing would solve this issue. Similarly, "cytocenter" is a flashy terminology, but not properly defined. Figure legends could be improved in terms of annotation of cells and vesicles counted, specifics related to standard errors which have been plotted, etc.

---

## [Decision Letter · Decision Letter 3]

11 Oct 2021

Dear Dr Cookson,

Thank you for submitting your revised Research Article entitled "Pathogenic mutations in LRRK2 sequester Rab8a to damaged lysosomes and regulate transferrin-mediated iron uptake in microglia" for publication in PLOS Biology. I have now obtained advice from original reviewer #2 and have discussed their comments with the Academic Editor. 

Based on the review, we are very positive about your manuscript. However, before we can make a decision about publication, we would like you to address the reviewer's lingering concerns, paying special attention to points 1 and 2. We will then assess your revised manuscript and your response to the reviewer's comments, and we may consult the reviewers again.

Please also make sure to address the following data and other policy-related requests.

1) Title: We suggest the following title, which we think might be more appealing to a broad readership: "Mutations in LRRK2 linked to Parkinson’s disease sequester Rab8a to damaged lysosomes and regulate transferrin-mediated iron uptake in microglia." However, we would be happy to work with you on an alternative, if our recommendation misrepresents your findings.

2) Blurb: Please provide a blurb which (if accepted) will be included in our weekly and monthly Electronic Table of Contents, sent out to readers of PLOS Biology, and may be used to promote your article in social media. The blurb should be about 30-40 words long and is subject to editorial changes. It should, without exaggeration, entice people to read your manuscript. It should not be redundant with the title and should not contain acronyms or abbreviations. For examples, view our author guidelines: https://journals.plos.org/plosbiology/s/revising-your-manuscript#loc-blurb

3) Data not shown: Please note that per journal policy, we do not allow the mention of "data not shown", "personal communication", "manuscript in preparation" or other references to data that is not publicly available or contained within this manuscript. Please either remove mention of these data or provide figures presenting the results and the data underlying the figure(s).

4) Blot and gel reporting requirements: We require the original, uncropped and minimally adjusted images supporting all blot and gel results reported in an article's figures or Supporting Information files. We will require these files before a manuscript can be accepted so please prepare and upload them now. Please carefully read our guidelines for how to prepare and upload this data: https://journals.plos.org/plosbiology/s/figures#loc-blot-and-gel-reporting-requirements

5) Ethics:

5.a) Please include within your manuscript the ID number of your protocols approved by the Institutional Animal Care and Use Committee of National Institute on Aging, NIH.

5.b) Please include the specific national or international regulations/guidelines to which your animal care and use protocol adhered. Please note that institutional or accreditation organization guidelines (such as AAALAC) do not meet this requirement.

6) Data: You may be aware of the PLOS Data Policy, which requires that all data be made available without restriction: http://journals.plos.org/plosbiology/s/data-availability. For more information, please also see this editorial: http://dx.doi.org/10.1371/journal.pbio.1001797

Note that we do not require all raw data. Rather, we ask for all individual quantitative observations that underlie the data summarized in the figures and results of your paper. For an example see here: http://www.plosbiology.org/article/info%3Adoi%2F10.1371%2Fjournal.pbio.1001908#s5

These data can be made available in one of the following forms:

I) Supplementary files (e.g., excel). Please ensure that all data files are uploaded as 'Supporting Information' and are invariably referred to (in the manuscript, figure legends, and the Description field when uploading your files) using the following format verbatim: S1 Data, S2 Data, etc. Multiple panels of a single or even several figures can be included as multiple sheets in one excel file that is saved using exactly the following convention: S1_Data.xlsx (using an underscore).

II) Deposition in a publicly available repository. Please also provide the accession code or a reviewer link so that we may view your data before publication.

Regardless of the method selected, please ensure that you provide the individual numerical values that underlie the summary data displayed in the following figure panels: Figures 1BCF, 2B, 3B, 4BDEFGJK, 5CDFGHI, 6BCDEF, 7C, 8BC, S2BC, S3BD, S4BC, S5BDE, S6BCE, and S7BD.

6.a) Please also ensure that each figure legend in your manuscript includes information on where the underlying data can be found and that your supplemental data file/s has/have a legend.

6.b) Please ensure that your Data Statement in the submission system accurately describes where your data can be found.

We expect to receive your revised manuscript within two weeks.

*Published Peer Review History*

*Early Version*

Sincerely,

Gabriel Gasque, Ph.D.,

Senior Editor,

ggasque@plos.org,

PLOS Biology

Reviewer remarks:

Reviewer #2: I am satisfied with the author´s rebuttal and revised version of the manuscript, and have only minor comments which need to be addressed as indicated below:

1. Figure 1 E: the colocalization of Rab8a with LRRK2-G2019S in the live cell image is convincing, but not the colocalization of these two markers with LAMP1. Please replace with another image where the colocalization of all three markers is more evident.

2. Figure 4: Panel J is confusing. The authors state that "Knockdown of Rab8a expression rescued the increase in the proportion of Tf that co-localized with LRRK2 relative to total internalized Tf caused by mutant LRRK2". Whilst there is a clear a difference in the numbers from ctrl versus Rab8a RNAi in I2020T expressing cells, it would be good to include data from wt expressing cells. Otherwise, rephrase to state that knockdown of Rab8a decreased the proportion of Tf that co-localized with LRRK2, rather than a statement regarding "rescue".

3. Results: "Lastly, ICP-MS analysis revealed elevated iron levels in cells stably-expressing G2019S LRRK2 compared to WT LRRK2". Please also state that the pathogenic R1441C and Y1699C mutants do not show detectable alterations in iron levels (Fig. 4K), at least as measured by ICP-MS.

4. The authors show increased colocalization of TfR with LAMP2 in pathogenic LRRK2-expressing cells (Figure S5A,B), and they show that LAMP2-positive structures are cathepsin D-negative (Figure 2A). Whilst their data support the notion that pathogenic LRRK2 causes TfR accumulation in damaged (degradation-deficient) lysosomes, it remains possible that the identity of the vesicular compartment where TfR accumulates is not lysosomal, but endosomal in nature (see eg. Cheng et al., JCB, 2018; 40% of LAMP1 can colocalize with the early endosomal marker EEA1 in the soma of cultured neurons). Perhaps a small mention in the discussion regarding this possibility is warranted.

5. Figure legend 1: (A)... stained for FLAG LRRK2, endogenous Rab8a and endogenous Lamp1. This should read Lamp2, not Lamp1.

6. Methods, section immunocytochemistry: please specify antibodies used against LAMP1, LAMP2 and cathepsin D.

---

## [Editor Report · Decision Letter 4]

1 Nov 2021

Dear Dr Cookson,

Thank you for submitting your revised Research Article entitled "Mutations in LRRK2 linked to Parkinson’s disease sequester Rab8a to damaged lysosomes and regulate transferrin-mediated iron uptake in microglia" for publication in PLOS Biology. I have now discussed this new version with other staff editors and with the Academic Editor, and I am pleased to tell you that we will probably accept this manuscript for publication, provided you satisfactorily address these remaining points:

1) The academic editor is now satisfied with the quality of the images that you have provided as well as with the other revisions. However, s/he notes that you should i) indicate the total number of experiments for each figures, and ii) more clearly state in all figure legends whether the “n” represents true replicates (independent experiments) or pseudo replicates (cells or images from a single/few experiments). For example, we understand that in figures 1B and C, “n” does not represent independent experiments but cells that could be obtained from one or few experiments.

2) Please also ensure that each figure legend in your manuscript includes information on where the underlying data can be found. You can write, for example, “The underlying data can be found in S1 Data/GEO accession numbers GSE186483 and GSE186559”

3) Please incorporate your “data not shown” into a supporting figure, updating the data files as well. There is at least one case of “data not shown”: “Mature iPSC-derived microglia were characterized by western blot and qPCR for expression of Iba1 protein and AIF1, TMEM119 and P2RY12 mRNA, respectively. (data not shown).”

4) Please update Figure S8 to include the raw image of the gels shown in Figure S1B. 

We expect to receive your revised manuscript within two weeks. 

*Published Peer Review History*

*Early Version*

Sincerely,

Gabriel Gasque, Ph.D.,

Senior Editor,

ggasque@plos.org,

PLOS Biology

---

## [Editor Report · Decision Letter 5]

10 Nov 2021

Dear Dr Cookson,

On behalf of my colleagues and the Academic Editor, Thomas C Südhof, I am pleased to say that we can in principle accept your Research Article "Mutations in LRRK2 linked to Parkinson’s disease sequester Rab8a to damaged lysosomes and regulate transferrin-mediated iron uptake in microglia" for publication in PLOS Biology, provided you address any remaining formatting and reporting issues. These will be detailed in an email that will follow this letter and that you will usually receive within 2-3 business days, during which time no action is required from you. Please note that we will not be able to formally accept your manuscript and schedule it for publication until you have any requested changes.

PRESS

Sincerely, 

Gabriel Gasque, Ph.D. 

Senior Editor 

PLOS Biology

ggasque@plos.org